# Azobenzene-based optoelectronic transistors for neurohybrid building blocks

Federica Corrado[1,2,3,10], Ugo Bruno[3,4,10], Mirko Prato [5], Antonio Carella[6], Valeria Criscuolo[1,2,3], Arianna Massaro [6], Michele Pavone [6], Ana B. Muñoz-García [7], Stiven Forti [8], Camilla Coletti [8], Ottavia Bettucci[3,9] ✉ & Francesca Santoro [1,2,3] ✉

Exploiting the light–matter interplay to realize advanced light responsive multimodal platforms is an emerging strategy to engineer bioinspired systems such as optoelectronic synaptic devices. However, existing neuroinspired optoelectronic devices rely on complex processing of hybrid materials which often do not exhibit the required features for biological interfacing such as biocompatibility and low Young's modulus. Recently, organic photoelectrochemical transistors (OPECTs) have paved the way towards multimodal devices that can better couple to biological systems benefiting from the characteristics of conjugated polymers. Neurohybrid OPECTs can be designed to optimally interface neuronal systems while resembling typical plasticity-driven processes to create more sophisticated integrated architectures between neuron and neuromorphic ends. Here, an innovative photo-switchable PEDOT:PSS was synthesized and successfully integrated into an OPECT. The OPECT device uses an azobenzene-based organic neuro-hybrid building block to mimic the retina's structure exhibiting the capability to emulate visual pathways. Moreover, dually operating the device with opto- and electrical functions, a light-dependent conditioning and extinction processes were achieved faithful mimicking synaptic neural functions such as short- and long-term plasticity.

Light-enabled bioelectronic devices represent an effective bridge between electronic and biological systems allowing several applications ranging from biodetection[1-4] to the complex biomimicry of biological pathways[5-10]. In fact, exploiting the use of light in combination with more conventional electronic-based methods enables a number of applications in engineering bioinspired systems such as retina-like visual devices[5-8,11] or synaptic neuromorphic platforms[9,10,12]. Here we demonstrated the capability of optoelectronic synaptic devices to mimic retinal visual pathways as well as short- and long-term plasticity[5-10].

Traditional inorganic and hybrid optoelectronic platforms find major limitations in biologically-driven systems where ion sensing as

[1]Institute of Biological Information Processing IBI-3 Bioelectronics, Forschungszentrum Juelich, 52428 Juelich, Germany. [2]Neuroelectronic Interfaces, Faculty of Electrical Engineering and IT, RWTH Aachen, 52074 Aachen, Germany. [3]Tissue Electronics, Center fo Advanced Biomaterials for Healthcare, Istituto Italiano di Tecnologia, 80125 Naples, Italy. [4]Dipartimento di Ingegneria Chimica, dei Materiali e della Produzione Industriale, Università degli Studi di Napoli Federico II, 80125 Naples, Italy. [5]Materials Characterization Facility, Istituto Italiano di Tecnologia, 16163 Genoa, Italy. [6]Dipartimento di Scienze Chimiche, Università degli Studi di Napoli "Federico II", Complesso Universitario Monte S. Angelo, 80126 Naples, Italy. [7]Dipartimento di Fisica "E. Pancini", Università degli Studi di Napoli "Federico II", Complesso Universitario Monte S. Angelo, 80126 Naples, Italy. [8]Center for Nanotechnology Innovation, Istituto Italiano di Tecnologia, 56127 Pisa, Italy. [9]Present address: Department of Materials Science and Milano-Bicocca Solar Energy Research Center – MIB-Solar, University of Milano-Bicocca, 20125 Milano, Italy. [10]These authors contributed equally: Federica Corrado, Ugo Bruno. ✉e-mail: ottavia.bettucci@unimib.it; f.santoro@fz-juelich.de

well as tailored biomechanical properties are required[7,11]. Therefore, to fabricate multimodal platforms capable of mimicking and potentially also directly interfacing with biological systems such as neuronal cells several strategies have been proposed for the development of optoelectronic devices featuring electro-mechanical stability in aqueous environment, ionic-electronic transduction and biophysical properties (i.e., biocompatibility, low stiffness[13–15]).

In this scenario, organic photoelectrochemical transistors (OPECTs) bearing light-responsive materials primarily at the gate terminal are experiencing substantial advances as bioelectronic platforms for biosensing and stimulation[1–4,16–19], whereas the integration of photosensitive hybrid composites is necessary to impart light-response capabilities to the devices. However, the synthetic processing of these materials require complex approaches[3,4,20], which potentially limits their use in more 'bio-compliant' architectures[2,3]. Considering the aforementioned challenges, engineering the integration of photo-switchable molecules into transistor architectures (e.g., spyropyrans and azobenzenes) already exploited in organic field effect transistors (OFETs) and organic thin film transistor (OTFTs), offers a promising strategy to fabricate fully organic OPECTs capable of both mimicking and interfacing biological systems. Here, we present an OPECT-based platform including a fully organic light-responsive gate electrode featuring a light-responsive poly(3,4-ethylenedioxythiophene) polystyrene sulfonate (PEDOT:PSS) covalently bonded to azobenzenes moieties (azo-tz-PEDOT:PSS) through click-chemistry. The integration of the azo-tz-PEDOT:PSS as a planar gate in a OPECT was also demonstrated to exhibit current modulation under different light stimulation conditions. Interestingly, this newly synthesised light-responsive conjugated polymer and its integration as gate electrode gives rise to an "all-in-one" device capable of both biomimicking the vertebrate retina's visual pathways and operating as an optical neuromorphic synaptic device able to emulate both short- and long- term plasticity (STP and LTP). Ultimately, the designed neuromorphic platform was also exploited to emulate learning mechanisms through dynamic light-induced synaptic conditioning and extinction. The versatility of high-performing fully organic building blocks paves the way to obtaining multimodal platforms capable of mimicking, sensing, and stimulating biological systems in an efficient, and cost-effective way.

## Results and discussion

Azo-tz-PEDOT:PSS was obtained by electrodeposition of the polymer precursor N₃-PEDOT and a 'click chemistry' reaction was designed to further functionalise the film via *Cu (I)-catalysed azide-alkyne Huisgen [3 + 2] cycloaddition* with the azo-alkyne monomer[21,22] (Fig. 1a, Methods section and Supplementary Figs. 1–5). The presence of the characteristic triazole ring at the azobenzene region was confirmed by Fourier Transform Infrared - Attenuated Total Reflectance (FTIR-ATR) and X-ray photoelectron spectroscopy (XPS) (Supplementary Figs. 6–8).

Furthermore, surface morphology characterisation highlighted the peculiar globular shape of the conjugated polymer with PEDOT and PSS randomly distributed domains (brighter and darker regions, respectively) also retained after the 'click' reaction. The surface functionalization only introduced a minor surface smoothening (Fig. 1b and Supplementary Fig. 9).

Azobenzenes might undergo *trans-cis* isomerization upon light irradiation at a characteristic wavelength: here, the UV-Visible spectrum indeed displayed an absorption peak at 347 nm, indicative of the *trans*-azobenzene isomer, and a less intense peak at 434 nm, corresponding to the *cis*-azobenzene (Fig. 1c, violet solid line). After light exposure ($t = 5$ min, $\lambda = 365$ nm, 6 W), a decrease in the absorption peak was observed as well as a slight increase of the peak at 434 nm[23] (Fig. 1c, violet dashed line).

Moreover, the reverse isomerization from *trans* to *cis* isomer was investigated exploiting the use of temperature ($T = 80°$) and blue light ($\lambda = 445$ nm, 440 mW) showing in both cases a complete recovery of the *trans* isomer but with different kinetics of the reaction (Supplementary Figs. 10 and 11).

In addition, cyclic voltammetry (CV) measurements showed two characteristic peaks: a reduction peak at −0.12 V that might be attributed to the reduction of the azobenzene N = N bond induced by the presence of protons from ⁻H₂PO₄ in the electrolyte solution, and an oxidation peak at 0.57 V (Fig. 1d, violet solid line) which might be due to the oxidation of the azobenzenes[24,25].

Upon light exposure, the CV hysteresis of the azo-tz-PEDOT: PSS film exhibits a further decrease, probably induced by the *trans-cis* photoisomerization (Fig. 1d, violet dashed line) where the bent shape of the *cis*-azobenzene creates an additional impediment to the charge flow, limiting the tunnelling effect promoted by the linear *trans*-isomer[26]. Complementarily, electrochemical impedance spectroscopy (EIS) highlighted a decrease in capacitance in the case of azo-tz-PEDOT:PSS when compared to pristine PEDOT:PSS (Fig. 1e), provided by the presence of the additional molecular layer of azobenzenes acting as a spacer between the electrolyte and the polymer surface (Supplementary Fig. 12 and Supplementary Table 1)[27].

Then, the azo-tz-PEDOT:PSS film was integrated as the gate electrode in a planar OPECT (Fig. 1a, Methods section and Supplementary Figs. 13 and 14) to build the so called azo-OPECT: transfer-characteristic curves showed the quasi-linear response of the drain current ($I_{ds}$) when the azo-tz-PEDOT:PSS gate voltage modulated the output current (Fig. 1f). Interestingly, upon light exposure ($t = 5$ min, $\lambda = 365$ nm, 6 W), the $I_{ds}$ current shifts to lower values suggesting a photo-induced charge transfer/charge trapping mechanism occurring between the azobenzenes and the PEDOT:PSS backbone (Fig. 1g), unlike pristine PEDOT:PSS which typically does not show photo-thermoelectrical behaviour[28] (experimental setup is reported in the Supplementary Fig. 15).

Density functional theory and time-dependent density-functional theory (DFT/TD-DFT) calculations were performed on an azo-tz-PEDOT model system to support the interpretation of the light-responsive mechanisms and to provide an unbiased energetic picture of the possible light-induced charge transfer processes (Supplementary Figs. 16–19) yet confirming the nature of the UV-vis transitions for *cis* and *trans* conformers (Supplementary Fig. 18 and Fig. 1c). For the *cis* conformer, we also predicted the following energy diagram: the HOMO is localised on the PEDOT backbone, the LUMO is instead localised on the azo-tz moiety; the other relevant MO is the HOMO-2 that is localised onto the azo-tz moiety (Fig. 1g). Within this energy framework, upon UV light irradiation, the photoexcited electron can be transferred from the LUMO (on the azo-tz) to the HOMO on the PEDOT backbone conductive system. Population of states located on the azobenzenes would lead to spatially separated charge carriers that, in principle, can be trapped on the azo side (positive charge highlighted in Fig. 1g). This mechanism would clarify the polarisation at the gate electrode and the resulting decrease in the current flow.

The role of light exposure was instrumental in addressing further controllable modulation of the channel current ($I_{ds}$), given its inverse relationship with the light intensity (100% = 2.8 W/cm², 60% = 1.7 W/cm², 20% = 0.56 W/cm², Fig. 2a, b).

Here, subsequent light pulses at the highest intensity (alternated by dark intervals) were applied at the gate electrode while no electrical bias was applied. Therefore, a photoexcitation process was triggered, resulting in the polarisation of the gate electrode (positive charges accumulation, Fig. 2c-i). The gating mechanism favoured positively charged ions to penetrate the bulk of the polymeric channel, causing the de-doping of the PEDOT:PSS channel and the resulting decrease of $I_{ds}$ (Fig. 2c-ii).

Here, the rapid ON-OFF behaviour confirms that the conductivity of the azo-tz-PEDOT:PSS gate is not only governed by the *trans-cis* isomerization but is combined with the photo-induced charge transfer effect occurring at the electrode surface. Previous

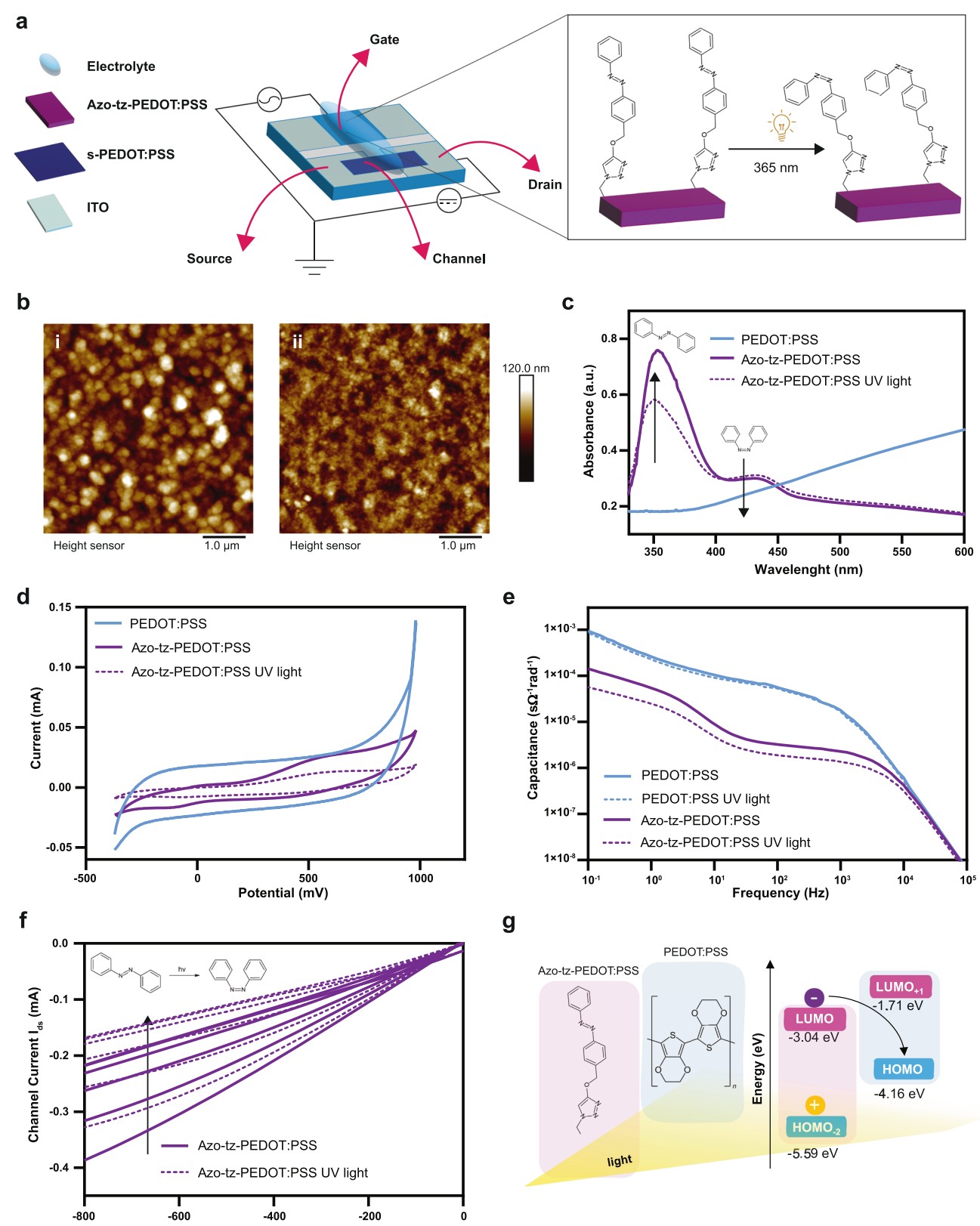

work carried out on similar azobenzene monomers showed how the spontaneous *cis-trans* relaxation occurs in the order of minutes[29], while here the current of the azo-OPECT reaches its initial value within 10–30 s, suggesting that the *cis-trans* isomerization contribution can only have an influence within the first light pulse[29,30].

However, the resulting charge at the gate electrode was constant for the first three pulses, hence the contribution of the *trans-cis* isomerization might be negligible (Fig. 2d and Supplementary Fig. 20). Furthermore, when the first light pulse is applied at the gate electrode, the charge density increases with the light intensity, confirming a significant photo-induced charge transfer mechanism (Fig. 2e and

**Fig. 1 | Synthesis and characterisation of azo-tz-PEDOT:PSS. a** Schematics of the proposed organic photoelectrochemical transistor featuring azo-tz-PEDOT:PSS planar gate (with *cis* photo-isomerization upon UV light irradiation) and spin coated PEDOT:PSS channel. **b** AFM images of PEDOT:PSS (i) and azo-tz-PEDOT:PSS (ii). **c** UV-visible spectra with *trans-cis* isomerization spectra before and after UV light stimulation (violet solid and dashed lines, respectively). **d** Cyclic voltammograms of PEDOT:PSS (blue solid line) and azo-tz-PEDOT:PSS, before and after 5 min of UV irradiation (violet solid and dashed lines, respectively): a slight reduction of the azo-tz-PEDOT:PSS hysteresis was observed compared to the pristine PEDOT:PSS which can be attributed to the chemical functionalization that involved the use of sodium ascorbate reducing the PEDOT:PSS film with a consequent a reduction in conductivity[29]. Both polymers showed a capacitive electrochemical behaviour in

agreement with literature[30]. **e** PEDOT:PSS capacitance before and after 5 min of UV irradiation (blue solid and dashed lines, respectively), azo-tz-PEDOT:PSS capacitance before and after 5 min of UV irradiation (violet solid and dashed lines respectively). **f** Transfer-characteristics curves of azo-OPECT before and after 5 min of UV irradiation (violet solid and dashed line, respectively) acquired by stepping the gate-source voltage ($V_{gs}$) in the range $-200$–$800\,mV$ (50 mV steps), while sweeping $V_{ds}$ in the range $-800$–$100\,mV$; **g** Chemical structure of the azo-tz-PEDOT system and energy levels of *cis* conformer obtained from DFT/TDDFT calculations. Only HOMO $-2$ /LUMO levels localised on azo-tz (pink box) and HOMO/LUMO $+1$ localised on PEDOT (blue box) are displayed. The arrow shows the possible photoinduced electron transfer from the LUMO level (localised on the photoexcited azobenzene) to the HOMO level of the system (localised on the PEDOT backbone).

Supplementary Fig. 20). Notably, while electrical gating induces a capacitive gate current, the light exposure generates a faradic gate current, as charges are transferred from the gate to the electrolyte, during the whole light exposition, without showing a significant decay (Fig. 2c-i). Such current generation agrees with the light-induced mechanisms previously proposed (Fig. 1g): upon light exposure, electrons are transferred from the LUMO (azo-tz) to the HOMO (PEDOT backbone), while holes are kept in the azo-tz. As a result, the gate/electrolyte interface potential decreases inducing a faradic current in the electrolyte, de facto increasing the channel/electrolyte potential, de-doping the polymeric channel[31].

Then, the azo-OPECT was then integrated into an electrical circuit to implement a biohybrid retinal architecture. Figure 2f–i shows the typical retinal cells' assembly into multilayers which also guide the vertical information pathway especially triggering ON/OFF states between photoreceptors and bipolar cells (BCs).

Typically, when photoreceptors are exposed to light, glutamate release decreases, inducing a conformational change in the OFF BCs receptors, with a consequent closure of the membrane pores. Conversely, when the concentration of glutamate increases, synaptic activation decreases, opening cation channels and depolarising the cell[32]. Finally, the inputs resulting from BCs are transmitted to ganglion cells (GCs) which are responsible for action potential (AP) generation. In the presence of glutamate, GCs are in a depolarised state (APs generation) while in the absence of glutamate they are hyperpolarized (no APs generation)[33].

In light of this, the retina cell layers were emulated by a connection in series of an azo-OPECT and a PEDOT:PSS-based resistor (Fig. 2f-ii, Methods). Here, the azo-tz-PEDOT:PSS gate acts as the photoreceptor layer, the electrolyte-channel interface mimics the BCs layer, while the PEDOT: PSS-based organic resistor acts as the GCs (Fig. 2f-ii,). In brief, the output voltage $V_{GC}$ is determined by the conductance of the azo-OPECT channel, which is controlled *via* the light input.

Interestingly, under dark conditions, the equivalent ganglion cell (output voltage $V_{GC}$) fires continuously, mimicking the continuous biological depolarisation mechanism induced by the glutamate release (vertical OFF pathway). The lateral pathway with a variable gain of the photoreceptors was emulated by the application of a voltage bias ($V_L$) at the gate terminal of the azo-OPECT. In fact, when $V_L$ was increased up to 300 mV (Fig. 2g-i, green box), the $V_{GC}$ decreased to 15% (Fig. 2g-ii, green box). This output voltage value was fixed as a firing threshold (Fig. 2g-ii, horizontal black solid line): whenever $V_{GC}$ crossed this threshold, an AP was released.

Furthermore, both vertical and lateral pathways can be mimicked upon the application of a light stimulus (Fig. 2g-iii). When the biohybrid retina is exposed to light, the equivalent GCs slowly stop firing (voltage drops by 18%), as the output voltage does not reach the firing threshold. Such process effectively resembles the OFF vertical pathway of the biological retina and the hyperpolarization of ganglion cells induced by the absence of glutamate upon light exposure. In this case,

a regulation was also possible by introducing the lateral pathway (300 mV) which further reduced the amplitude of $V_{GC}$ ($-20\%$), accounting for a total reduction of the output voltage of $-43\%$ (Fig. 2g-iii, green box).

Furthermore, we investigated how the azo-OPECT can also act as a building block in plasticity-driven biohybrid systems. As previously shown, this platform can mimic neuromorphic features when square voltage pulses are applied at the gate terminal, acting as action potentials at pre-synaptic ends in biological neuronal synapses and modulating the conductance at the channel which is considered as post-synaptic end (Fig. 3a)[34].

In our system, the pre-synaptic signal was a train of positive voltage pulses at the gate terminal ($V_{gs} = 300\,mV$, pulse width (PW) = 5 s) while $V_{ds}$ was fixed at $-200\,mV$ (Supplementary Fig. 21). When the light stimulus was applied ($\lambda = 365\,nm$, $t = 6\,min$), a decrease in the conductance of the post-synaptic channel was observed ($\Delta G\% = 12.5 \pm 3.1$; $N = 3$) (Fig. 3b, blue arrow). After removing the light stimulus, the channel conductance increased ($\Delta G\% = -2.7 \pm 3.1$; $N = 3$, Fig. 3b, red arrow) resulting in the conditioning of the synapse due to the photo-excitation process of the azo-tz-PEDOT:PSS. This induces a polarisation of the gate electrode whereas the memory recovery can be attributed to the diffusion of the cations trapped into the channel which return to the electrolyte. This light responsive behaviour of the neuromorphic azo-OPECT was further investigated by simultaneous variation of voltage pulse width (PW, Fig. 3c-i–iii) at the gate and the light intensity (Supplementary Fig. 22 and Supplementary Tables 2 and 3): in fact, the synaptic conditioning, *i.e.*, percentage variation of the post-synaptic channel conductance, increases transitioning from 20% to 60% light intensity independent of the applied PW (Fig. 3d) while a saturation was reached at 100% light intensity as discussed earlier (Fig. 2e).

In parallel, when the light intensity was kept fixed while varying the PW of the presynaptic input, no significant conductance difference was found, demonstrating that the light effect was independent of the duration of the electrical pulse (Supplementary Fig. 23 and Supplementary Table 2 and 3).

As light eventually induces a memory effect at the post-synaptic end, we investigated a possible mechanism driving different time-scale potentiation of the biohybrid synapse. As synaptic plasticity can be divided into short-term plasticity (STP), which lasts from milliseconds to seconds and represents the unitary information transmission or learning proves, and long-term plasticity (LTP) which includes synaptic weight changes lasting over longer time and is fundamental to learning and memory in neural system[29,35]. Here, paired pulse facilitation (PPF) was evaluated, in which the application of two consecutive pre-synaptic stimuli results in the enhancement of the amplitude of the second evoked post-synaptic potential[36]. Therefore, two electrical pulses with different time delays $\Delta t$ were applied at the gate terminal while measuring the elicited channel current spikes (Supplementary Fig. 24). A PPF index was then computed as the percentage increase in the amplitude of the second current spike, with respect to the first spike. When the light stimulus was applied, the postsynaptic current

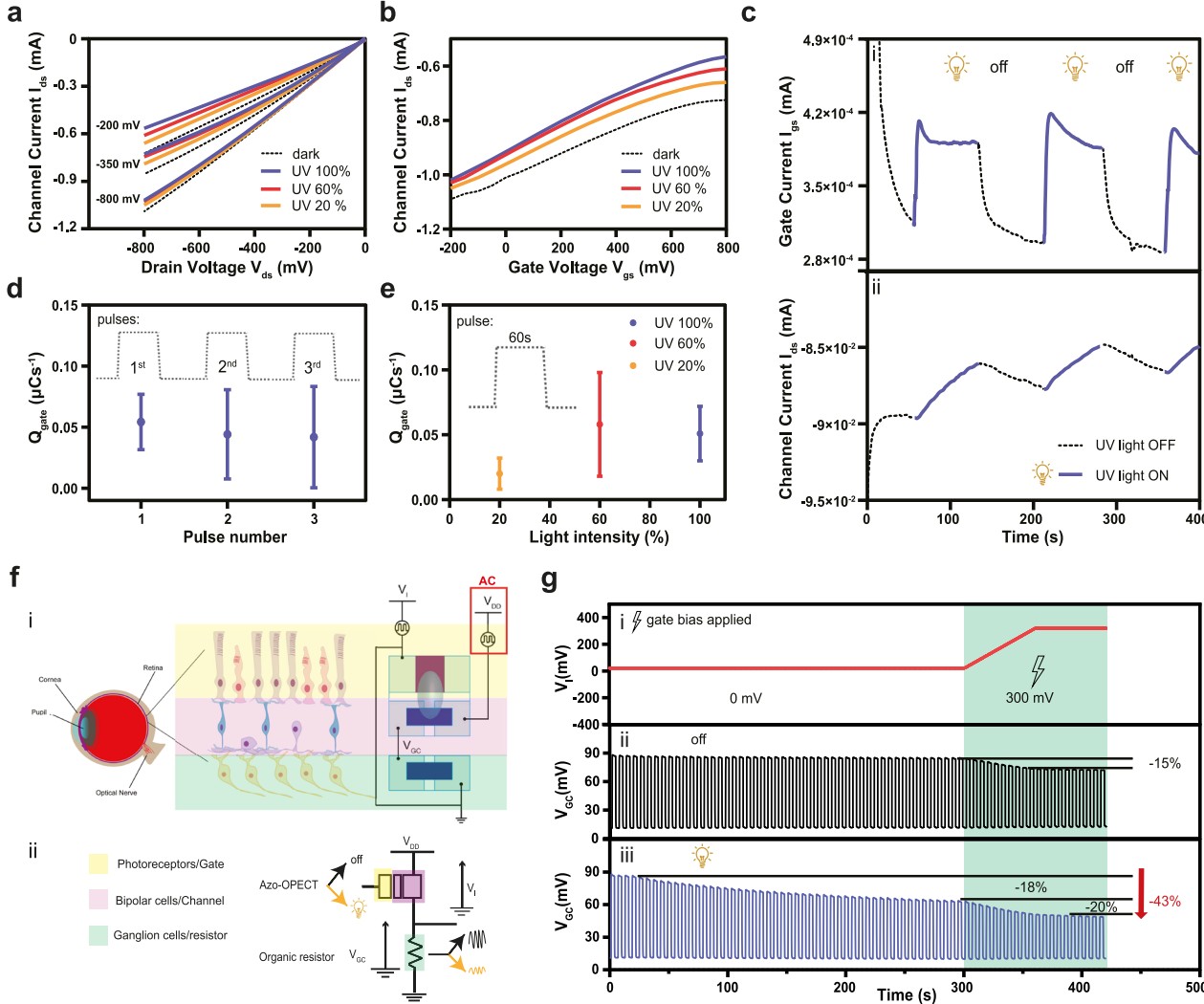

**Fig. 2 | OPECT operation under light conditions and retina visual pathway mimicking. a, b** Transfer-characteristic and output curves at different light intensity (100%, 60%, 20% blue, red and orange solid line respectively). **c** Gate and channel currents measured while alternating 60 s of dark condition (black dashed line) and 60 s of UV light (black solid line), $V_{ds} = -200$ mV and $V_{gs} = 0$ V: (i) a rapid increase in the $I_{gs}$ (from 310 nA to 414 nA) was observed (blue solid line) followed by an immediate decrease with a complete recover of the initial values after removing the light stimulus, (black dotted solid line), while (ii) shows a rapid decrease of $I_{ds}$ observed during light stimulation (blue solid line) while a rapid increase occurred when the stimulus was removed (black dotted line). **d** Normalised charge at the gate calculated by integrating $I_{gs}$ current vs. time (light intensity 100%) (numerical values: $0.054 \pm 0.023$ μC/s, $0.044 \pm 0.036$ μC/s and $0.042 \pm 0.042$ μC/s, $N = 5$). **e** Normalised charge at the gate calculated on the first pulse by integrating $I_{gs}$ current vs. time during UV irradiation of azo-tz-PEDOT:PSS at different light

intensity (20%, 60% and 100%) (Numerical Values: $0.020 \pm 0.012$ μC/s, $0.058 \pm 0.040$ μC/s, $0.051 \pm 0.022$ μC/s, $N = 3$). **f** Retina schematics and visual pathways. (i): OFF vertical pathways responsible to stream the information that drives light to the brain involving three main cell layers: the signal flows from the photoreceptors to bipolar cells (BCs) and from BCs to ganglion cells (GCs). The secondary lateral pathway adjusts the gain of the pre-synaptic and post-synaptic involving horizontal cells (HCs) back to photoreceptors and from amacrine cells (ACs) to BCs[34]. In dark conditions, photoreceptors are in depolarised state, and they continuously release glutamate. Contrariwise, the light induces hyperpolarization; (ii) proposed equivalent circuit which emulated the retina behaviour. **g** Artificial retina lateral and vertical OFF pathway emulation: (i) retina lateral pathway where the applied pulse was ramped up from 0 to 300 mV; (ii, iii) OFF vertical pathway: electrical train pulses application both in dark and light conditions, respectively.

was potentiated (PPF index ≤ 99%) if the second stimulus occurred within 5 s from the application of the first stimulus (Fig. 3e, blue solid trace).

On the other hand, under dark condition the potentiation was observed only in the case of a time distance between pulses of <2 s (Fig. 3e, black dotted trace). Finally, PPF indexes calculated under dark and light conditions are comparable, suggesting that STP is independent of the light exposure and mainly induced by the electrical stimulation (Supplementary Table 4).

When 500 light pulses were applied, each pulse caused a decrease in the channel conductance (Fig. 3f, inset) while after removing the

light stimulus, a slow increase in the channel conductance was observed which partially recovered the initial conductance level. After more than 30 min, the synaptic recovery was still partial, confirming the LTP of the neuromorphic azo-OPECT (Fig. 3f).

Given that a positive gate bias plays a key role in the synaptic conditioning (memory 'writing') of the neuromorphic azo-OPECT particularly enhanced with light stimulation, we investigated how the synapse could also provide an extinction behaviour (memory 'erasing') eventually by applying a negative bias voltage at the gate electrode.

First, 500 light pulses were applied at the gate, inducing a channel current decrease, and then a negative electrical pulse

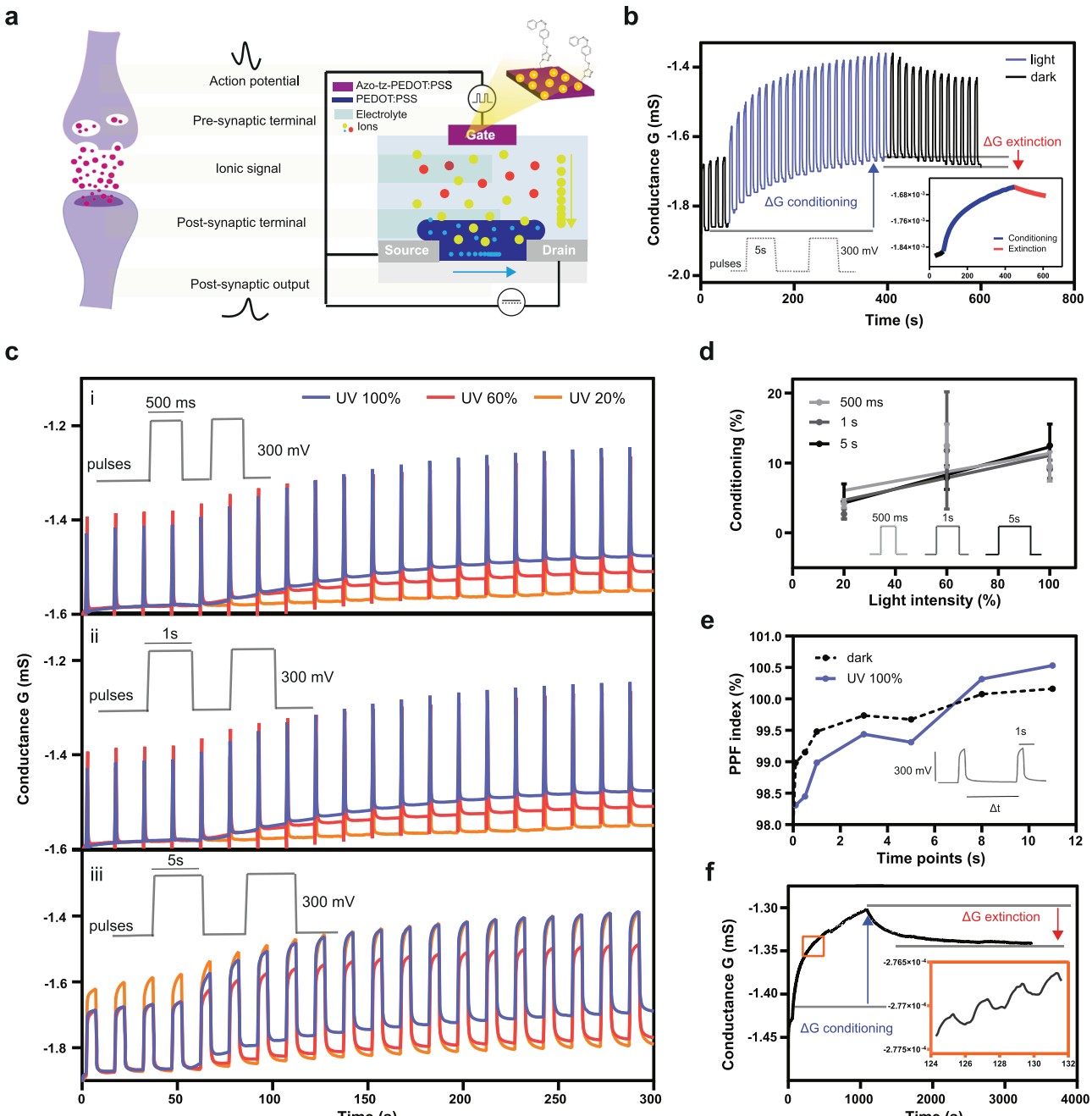

**Fig. 3 | Synaptic plasticity emulation upon light stimuli. a** Schematics comparing an OPECT with a biological synapse: when a square voltage pulse was applied at the gate, it acted as an action potential which reached the pre-synaptic terminal. This voltage pulse favoured the injection of cations from the electrolyte solution to the PEDOT:PSS channel in a similar manner of the chemical transduction of the information occurring during the neurotransmitters release from the pre-synaptic terminal. This signalling resulted in the ionic current at the post-synaptic membrane with a consequent conductance variation and, thus, a modulation of the $I_{ds}$ current can be recorded on the PEDOT:PSS channel of the OPECT. **b** Channel conductance measurement while electrical pulses were applied at the gate and light stimulus was not applied and then light was switched on (black and blue solid lines, respectively). Inset: light induced conditioning (blue line) and extinction in dark (red line). **c** Channel conductance measured at different light intensity (100, 60 and 20% blue, red and orange solid lines, respectively) by changing the PW: (i) 500 ms, (ii) 1 s and (iii) 5 s. **d** Channel conductance variation (conditioning) depending on PW of gate voltage pulses (PW: 500 ms, numerical values: $4.2 \pm 0.6$, $12.5 \pm 3.1$, $9.5 \pm 2.1$; PW: 1 s, numerical values: $2.7 \pm 0.7$, $11.8 \pm 8.4$, $9.1 \pm 1.3$; PW: 5 s, numerical values: $4.5 \pm 2.5$, $7.9 \pm 1.7$, $12.5 \pm 3.1$). **e** PPF index (Supporting Fig. 24) plotted vs. $\Delta t$; the measurement was carried out by applying two consecutive square voltage pulses with $V_{gs} = 300$ mV, PW = 1 s, $\Delta t = 100$ ms, 500 ms, 1 s, 3 s, 5 s, 8 s and 11 s). **f** LTP of azo-OPECT showing 500 stable conductance states and long-term memory after the irradiation with 500 light pulses ($\lambda = 365$ nm, 100% of light intensity, 1 s light on, 1 s light off); the conductance remained constant after the removal of the light stimulus.

($V_{gs} = -300$ mV, total duration 10 min) triggered the post-synaptic current to increase (Fig. 4a and Supplementary Fig. 25). Finally, by removing the electrical stimulus, the current values decreased again bringing the synaptic conditioning back to the initial values. As a control, transistors with a pristine PEDOT:PSS gate did not exhibit

any doping/de-doping effect under the same gate bias conditions (Supplementary Fig. 26).

To further investigate how ion migration might be induced by the negative voltage bias at the azo-tz-PEDOT:PSS gate electrode, a series of consecutive pulses of increasing intensity (from −20 mV to

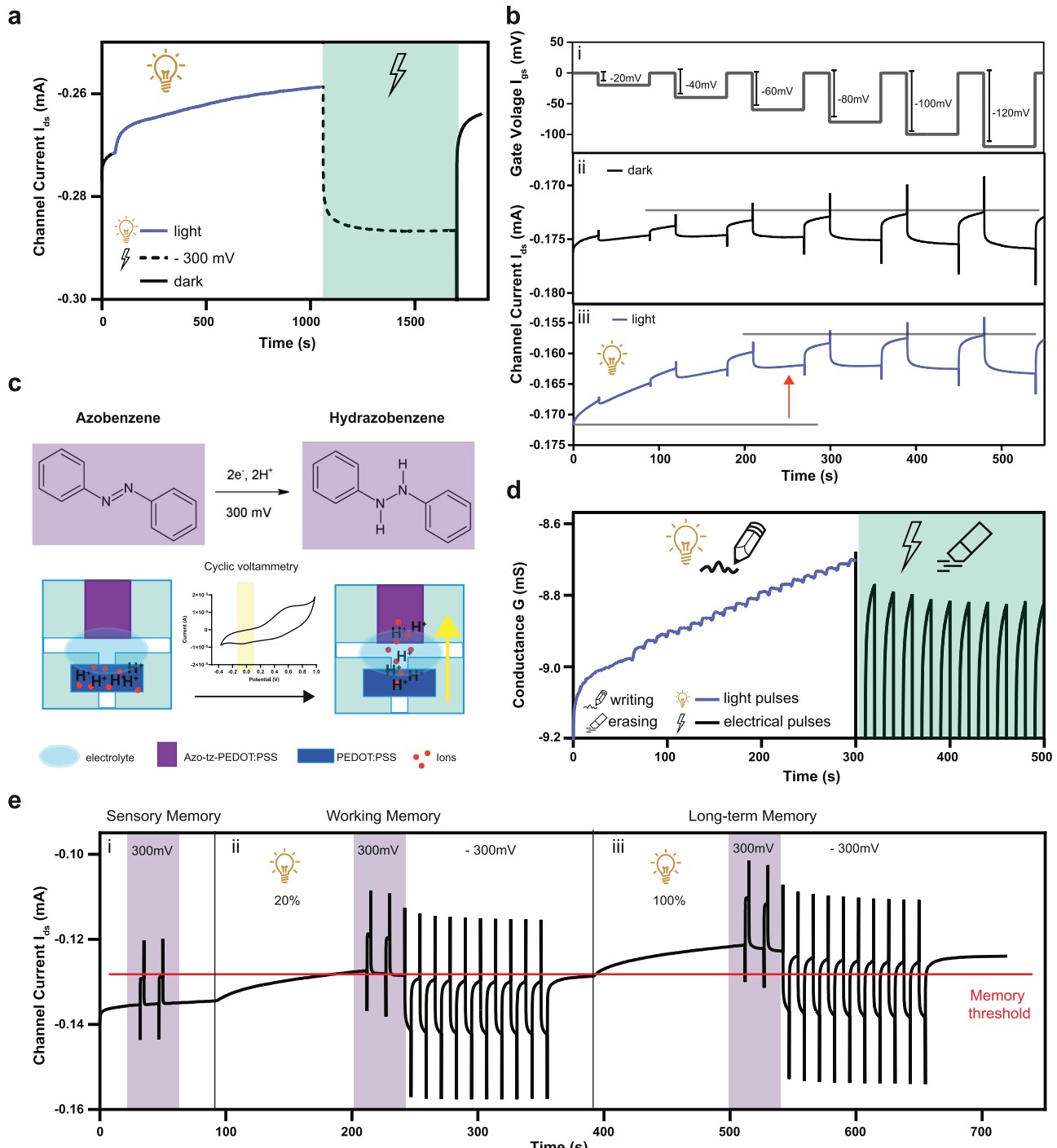

**Fig. 4 | Optoelectronic memory behaviour and neuromorphic applications.**
**a** Channel current recorded upon the application of a light stimulus (500 light pulses, $\lambda = 365$ nm, PW = 1 s, $\Delta t = 1$ s) (blue solid line) and a negative voltage pulse (black dashed line, green box). **b** Series of consecutive negative voltage pulses of increasing intensity from (i) $V_{gs} = -20$ mV to $V_{gs} = -120$ mV and resulting channel current measurements under (ii) dark and (iii) light conditions. **c** Schematics representing the mechanism of protons migration from the channel to the electrolyte towards the azo-tz-PEDOT:PSS gate terminal. **d** Operation of the azo-OPECT as optoelectronic memory (channel conductance measurement): write/erase processes induced by the application of light pulses train (20 light pulses, $\lambda = 365$ nm, PW = 2 s, $\Delta t = 10$ s) (blue solid line) followed by a negative square voltage pulses

train ($V_{gs} = -300$ mV, PW = 3 s, $\Delta t = 10$ s) (black solid line, green box). **e** Atkinson-Shiffrin memory model: (i) sensory memory: the sensory information (purple boxes) was rapidly forgotten. The $I_{ds}$ did not reach the memory threshold (horizontal red dotted line), (ii) working memory: a low intensity of light (20%) was applied. Initially, the current overcame the memory threshold. The short-term behaviour was emulated by applying negative square voltage pulses at the gate terminal. As a result, the current value returned below the threshold. (iii) Long-term memory: the high intensity of light (100%), emulating the rehearsal process, induced a current decrease which cross the threshold even when the negative voltage pulses were applied achieving the long-term memory.

−120 mV) were employed (Fig. 4b–i). Here, under dark conditions the post-synaptic current undergoes a significant increase only when the voltage pulse amplitude reaches −120 mV (Fig. 4b-ii and Supporting Information S12). When a constant light stimulus was applied, $I_{ds}$ decreased within the first two electrical pulses (at −20 mV and −40 mV) due to the gate electrode polarisation, transitioning to higher values when the gate bias amplitude was at least −80 mV (Fig. 4b-iii and Supplementary Fig. 27). For voltage values lower than −80 mV, the light stimulation elicits a modulation of the channel current while for higher gate voltage amplitudes, the electrical stimulation has a predominant effect on the channel current amplitude increase.

Here, the electrochemical reaction that reduces the azobenzenes to hydrazobenzene requires two protons and occurs at the gate electrode when the voltage applied is between −20 mV and −350 mV (Fig. 1d). Therefore, the negatively charged azobenzenes can attract protons from the channel through the electrolyte because of charge balancing, while on the other hand, the PSS⁻ counterions, stripped of a proton, restore the interaction with PEDOT⁺ with a consequent increase of the channel conductance (Fig. 4c).

To further corroborate this, we used a different electrolyte (i.e.,100 mM NaCl) containing two orders of magnitude less protons than PBS. In this case, the application of consecutive negative voltage pulses induced a low modulation of $I_{ds}$ confirming that the number of protons available in the electrolyte as an effect on the channel current modulation[37] (Supplementary Fig. 28).

Given the light- and voltage-induced antagonistic mechanisms, a synaptic optoelectronic memory was realised, in which the information can be written and erased by means of light and voltage inputs, respectively. In fact, "writing" operations were attained by applying a train of 20 light pulses at the gate electrode. Here, each single pulse writes a new information in the azo-OPECT that results in a distinct and non-volatile conductive state (Fig. 4d and Supplementary Fig. 29). Then, negative square voltage pulses were applied, resulting in a conductance decrease, erasing what written by the light (Fig. 4d, black solid line, green box).

In addition, these distinct addressable states of the neuromorphic azo-OPECT were further exploited to emulate complex learning paradigms such as the Atkinson-Shiffrin memory model[38]. According to this classical multi-store model, memory is organised in three separate components: sensory, short- and long-term memory. When an external stimulus is perceived, a sensory memory is formed, which remains in the sensory register for a very brief period before being forgotten. When attention is given to the information provided by the sensory input, the memory can become a short-term memory. Moreover, through rehearsal, a short-term memory can become a long-term memory[38]. Here, Fig. 4e shows how the Atkinson-Shiffrin model might be replicated with sensory information input (positive pre-synaptic input) that was rapidly forgotten when there was no rehearsal process (purple boxes), and the post-synaptic channel current did not reach the memory threshold (Fig. 4e-i). To emulate attention, a low intensity light stimulus (20%) was applied, which shifted the information coming from sensors to short-term memory as the memory threshold was exceeded. However, such sensory information was forgotten after the application of a negative pre-synaptic input, emulating the process of forgetting, bringing the current values below the memory threshold (Fig. 4e-ii). To emulate the transition of sensory memory from short- into long-term memory through rehearsal, a high intensity light stimulation (100%) was applied. The resulting sensory memory is not erased by a negative pre-synaptic input; thus, the current was above the memory threshold, proving the long-term memory of the device (Fig. 4e-iii).

In summary, in this work an innovative and versatile light-sensitive PEDOT:PSS covalently bonded to azobenzene moieties was synthesised and characterised. A highly reproducible surface functionalization at the sub-nanoscale of an electrodeposited PEDOT:PSS film was performed while a deep morphological, chemical, and electrical characterisation of the azo-tz-PEDOT:PSS demonstrated the possibility of modulation of its electrical properties under different UV light stimuli. The optoelectronic response of azo-tz-PEDOT:PSS allowed its integration as a light responsive gate in an OPECT architecture which was successfully achieved, showing unprecedented light-induced features that allow its use as a neurohybrid building block in several bioelectronics applications. The possibility of output current tuning depending on the light intensity was exploited to mimic both the lateral pathway and the OFF vertical retina visual pathways. Additionally, by optimising the geometrical parameters of the OPECT channel, the light-mediated gating efficiency could be finely tuned, also reducing the operation speed of the device.

Moreover, combining light and electrical stimuli, the azo-OPECT exhibited neuromorphic features in processing of information like a biological neuron, mimicking the synaptic information transmission such as STP and LTP. Additionally, the controllable conditioning and extinction processes induced by light and electrical stimuli were exploited to operate the azo-OPECT as an optoelectronic synaptic memory capable of write/erase also allowing emulation of the inner workings of human memory through the Atkinson-Shiffrin memory model.

## Methods
### Synthetic procedures
**2-Azidomethyl-2,3-dihydrothieno[3,4-b]−1,4-dioxine (EDOT-N₃).** 2-chloromethyl-2,3-dihydrothieno[3,4-b]−1,4-dioxine, EDOT-Cl (Merck Life Science S.r.l., Italy) (50 mg, 0.262 mmol, 1 eq) was dissolved in 3 mL of *N,N*-dimethylformamide (DMF) (Merck Life Science S.r.l., Italy) and stirred at r.t. under inert atmosphere until complete dissolution of the product. After the addition of sodium azide (NaN₃) (TCI chemicals, Belgium) (34 mg, 0.524 mmol, 2 eq), the reaction mixture was refluxed for 3 h. The reaction mixture was extracted with ethyl acetate (EtOAc) (Merck Life Science S.r.l., Italy) and the resulting combined organic layers were washed with water to remove DMF (Merck Life Science S.r.l., Italy). The obtained crude product was dried over sodium sulfate (Na₂SO₄) (Merck Life Science S.r.l., Italy) and concentrated under vacuum giving the EDOT-N₃ as a colourless oil (48 mg, 0.243 mmol, 93%).

$^1$H NMR (CDCl₃, 400 MHz, δH, ppm): 3.56 (ddd, 2H, CH₃-N₃); 4.07, 4.21 (dd, 2H, O-CH₂), 4.35 (m, 1H, O-CH), 6.40 (dd, 2H, S-CH). FTIR-ATR: ν (cm⁻¹): 3114, 2977, 2926, 2876, 2100, 1485, 1186, 1022, 765. The above $^1$H NMR and FTIR-ATR analytical data agreed with those reported in the literature[22].

**4-Propargyloxyazobenzene (azo-alkyne).** 4-(Phenylazo)phenol (TCI chemicals, Belgium) (50 mg, 0.252 mmol, 1 eq) was dissolved in previously degassed acetone (3 mL) (Merck Life Science S.r.l., Italy). Potassium carbonate (K₂CO₃) (Thermo Fisher Scientific Inc., Germany) (174 mg, 1.26 mmol, 5 eq) was added to the organic solution and the reaction mixture was stirred at r.t. under inert atmosphere for 90 min. Then, propargyl bromide (TCI chemicals, Belgium) (150 mg, 1.26 mmol, 5 eq) was added, and the reaction mixture was refluxed overnight, under inert atmosphere. Removal of the acetone under vacuum provided the solid crude product. It was then re-dissolved and extracted with EtOAc (Merck Life Science S.r.l., Italy) and the combined organic solution was washed with water and dried over Na₂SO₄ (Merck Life Science S.r.l., Italy). Removal of the solvent under vacuum was employed to yield the alkyne as an orange powder (56.5 mg, 0.239 mmol, 95%).

$^1$H NMR (CDCl₃, 400 MHz, δH, ppm): 7.94 (m, 2H, H-2), 7.89 (dt, 2H, H-2'), 7.54−7.42 (m, 3H, H-3', H-4'), 7.12 (m, 2H, H-3), 4.78 (d, 2H, CH₂), 2.57 (t, 1H, triple bond C−H). FTIR-ATR: ν (cm⁻¹): 3266, 3076, 2927, 2129, 1600, 1584, 1489, 1237, 1144, 1016.

The above $^1$H NMR and FTIR-ATR analytical data agreed with those reported in the literature[39].

**Electrodeposition procedure.** The electrochemical polymerisations were carried out by electrodeposition method with Autolab PGSTAT302N (Metrohm Italiana S.r.l, Italy) potentiostat/galvanostat interfaced with a personal computer, equipped with NOVA 2.1 software, through a potentiodynamical cyclic voltammetry (CV) (in the range of 0 V, +1.0 V and 0 V, +1.5 V for PEDOT and N$_3$-PEDOT, respectively) from a suspension of 7.5 mg mL$^{-1}$ of poly(sodium 4-styrenesulfonate) (PSSNa) (Merck KGaA, Germany) aqueous suspension containing monomer (0.1 M of EDOT (Merck KGaA, Germany) and 0.01 M of N$_3$-EDOT, respectively) at a scan rate of 50 mV s$^{-1}$ for ten cycles in order to obtain electrodeposited PEDOT:PSS and N$_3$-PEDOT:PSS (named ePEDOT:PSS and eN$_3$-PEDOT:PSS, respectively) films. The electrodeposition of films was carried out by using a patterned square ITO glass slide (25 mm×25 mm, Kintec, Hong Kong) as working electrode (WE), Ag/AgCl NaCl (3 M) (Redox.me, Sweden) as reference electrode (RE) and a platinum wire (Merck KGaA, Germany) as counter electrode (CE).

ITO electrodes were previously cleaned in an ultrasonic bath for 10 min in each of the following solvents: Alconox® detergent solution (Merck Life Science S.r.l., Italy), deionized water, acetone (Merck Life Science S.r.l., Italy), and 2-propanol (Merck Life Science S.r.l., Italy) and dried with compressed air.

Prior to the application of the potential, the aqueous suspension was vigorously stirred for 30 min and preserved in the fridge overnight. After the electrodeposition, the films were washed with water to remove both the PSSNa (Merck KGaA, Germany) and the excess of unreacted monomer and finally dried on hot plate at 120 °C for 1 h.

**Post-functionalization of eN$_3$-PEDOT:PSS via click chemistry.** The previously synthesised alkyne (141 mg, 0.6 mmol, 1 eq) was dissolved in tetrahydrofuran (THF) (Merck Life Science S.r.l., Italy) (30 mL), before the addition of 30 mL of an aqueous solution of CuSO$_4$•5H$_2$O (Merck Life Science S.r.l., Italy; 150 mg, 0.64 mmol, 1.07 eq) and sodium ascorbate (Merck Life Science S.r.l., Italy; 118.9 mg 0.53 mmol, 0.89 eq). The previous electropolymerized N$_3$-PEDOT:PSS film was then put in the resulting mixture and maintained under gentle stirring for 24 h. The obtained functionalized film was repeatedly washed with deionized water to remove the excess of catalyst and with THF to remove the excess of unreacted alkyne[40].

**Organic photoelectrochemical transistor fabrication.** Customised patterned ITO-coated glass (surface resistivity 20 Ω cm$^{-2}$, 25 mm × 25 mm, Kintec, Hong Kong, see also Fig. S7.2) were cleaned in an ultrasonic bath with Alconox® detergent solution (Merck Life Science S.r.l., Italy), deionized water, acetone (Merck Life Science S.r.l., Italy), and 2-propanol (Merck Life Science S.r.l., Italy; 10 min for each solvent) and dried with compressed air. Then, N$_3$-PEDOT:PSS was electrodeposited on the rectangular part of the ITO-coated glass according to the previously reported procedure. The resulting polymer film was then post-functionalized through a click reaction, submerging the film in a solution of THF:H$_2$O (1:1) in presence of a stoichiometric amount of the azo-alkyne (10 mM, 141 mg, 0.6 mmol, 1 eq), CuSO$_4$ • 5H$_2$O (Merck Life Science S.r.l., Italy) and sodium ascorbate (Merck Life Science S.r.l., Italy) for 24 h[40]. After several washes with deionized water and THF, the planar gate of azo-tz-PEDOT:PSS was obtained. After the functionalization, the film was covered with a PDMS mask (23 mm × 12 mm) prepared by following a literature procedure[34], poured in a mould and cured for 1 h. Then, the remaining part of the ITO-coated glass underwent a O$_2$ plasma activation (2 min, 20 W) to favour the spin coating of a previously prepared PEDOT:PSS aqueous solution (5 vol.% ethylene glycol (Merck Life Science S.r.l., Italy), 0.002 vol.% DBSA (Merck Life Science S.r.l., Italy), and 1 vol.% GOPS (Merck Life Science S.r.l., Italy) added to a commercial Hereaus Clevios, PH1000 (Hereaus group, Germany) that was sonicated

30 min before use. The PEDOT:PSS was spin coated at 2000 rpm. for 2 min with an acceleration of 400 rpm s$^{-1}$ and annealed 1 h at 140 °C. Finally, to remove the excess of PEDOT:PSS and pattern the OPECT channel, O$_2$ plasma dry etching (15 min, 100 W) was performed including a hard PDMS mask (15 mm × 7 mm, 23 mm × 12 mm) to cover both the planar gate and the channel with two PDMS masks. The so obtained OPECTs have a channel area of 15 mm × 7 mm, a gate area of 13 mm × 10 mm of which the illuminated area is 20 mm$^2$. Before the use, the device was gently rinsed in deionized water overnight to promote the PEDOT:PSS swelling.

**Organic resistor.** A two-squared patterned ITO-coated glass (Surface resistivity 20 Ω cm$^{-2}$, 25 mm × 12.5 mm, Xinyan Technology Ltd., Hong Kong) was cleaned with deionized water, acetone (Merck Life Science S.r.l., Italy), and 2-propanol (Merck Life Science S.r.l., Italy) (10 min for each solvent) and dried with compressed air. Then, the ITO-coated glass underwent a O$_2$ plasma activation (2 min, 20 W) to favour the spin coating of a previously prepared PEDOT:PSS aqueous solution (5 vol.% ethylene glycol (Merck Life Science S.r.l., Italy), 0.002 vol.% DBSA (Merck Life Science S.r.l., Italy), and 1 vol.% GOPS (Merck Life Science S.r.l., Italy) added to a commercial Hereaus Clevios, PH1000 (Hereaus group, Germany) that was sonicated 30 min before use. The PEDOT:PSS was spin coated at 2000 rpm for 2 min with an acceleration of 400 rpm s$^{-1}$ and annealed 1 h at 140 °C. Finally, to remove the excess of PEDOT:PSS and fabricate the organic resistor, a O$_2$ plasma dry etching (15 min, 100 W) was performed covering the PEDOT strip with a PDMS mask (15 mm × 7 mm). Before the use, the resistor was gently rinsed in deionized water overnight to promote the PEDOT:PSS swelling.

## Chemical characterisation

**Nuclear magnetic resonance spectroscopy.** In this work, $^1$H-NMR spectra were detected at 400 MHz with an Avance III instrument (Bruker, Germany). The samples were dissolved in *d*-chloroform (CDCl$_3$) at 30 mg mL$^{-1}$. Chemical shifts refer to the residual solvent peak (CHCl$_3$, δ = 7.26). Fourier transform infrared - attenuated total reflection spectroscopy (FTIR-ATR). FTIR-ATR spectra were recorded under ambient atmosphere in the range 4000–600 cm$^{-1}$, using a NICOLET 6700 FT-IR spectrometer (Thermo Fisher Scientific, Germany) interfaced with a personal computer equipped with OMNIC software suite. A total of 64 scans for each measurement with a resolution of 4 cm$^{-1}$ were used to average the absorbance/transmittance signal and reduce the background noise. Ultraviolet-visible spectroscopy (UV-Vis). UV-Vis spectra were acquired in transmission in the range 900–300 nm with a resolution of 0.4 nm, using a Cary 100 UV-visible spectrophotometer (Agilent Technologies, USA) interfaced with a personal computer equipped with the CARY WIN-UV software.

**X-ray photoelectron spectroscopy.** XPS analyses were carried out using an Axis Ultra DLD (Kratos analytical, Shimadzu group company, UK) spectrometer. Data were acquired using a monochromatic Al Kα source, operated at 20 mA and 15 kV. Wide scans were acquired at pass energy of 160 eV and energy step of 1 eV; high resolution spectra were acquired at pass energy of 10 eV, energy step of 0.1 eV. In both cases, the take-off angle was set at 0 degrees with respect to sample normal direction. All the analyses were carried out with an analysis area of 300 × 700 μm. The charge neutraliser system (Kratos analytical, Shimadzu group company, UK) was used on all specimens. Spectra were charge-corrected to the main line of the carbon 1 s spectrum (C-C bonds) set to 284.8 eV.

## Morphological characterisation

**Atomic force microscopy.** A Dimension Icon® AFM (Bruker Corporation, USA) with Scan Assyst, operating in tapping mode (the cantilever vibrates near its resonance frequency causing the tip to

oscillate up and down), was used in dry conditions and under ambient atmosphere to measure films surface topography and roughness. The cantilevers (Bruker, Model: SCANASYST-FLUID, spring constant 0.7 N m$^{-1}$, resonance frequency 150 Hz, Thickness = 600 nm, Width = 10 nm, Length = 70 nm). The scanned area was $5 \times 5 \, \mu m^2$ with a scan rate of 2 Hz. Nanoscope 2.0 software was used to evaluate the root mean square (RMS or $R_q$) roughness for each acquired area.

**Scanning electron microscopy.** Scanning electron micrographs of synthesised polymers were acquired sputtering a thin layer of Au (20 nm) on the films using a sputter coater (CRESSINGTON HR208). A field emission scanning electron microscopy (FESEM) ULTRAPLUS, (Zeiss Company, Germany) equipped with OXFORD detector for EDX (EHT = 10.00 kV, aperture size = 30.00 μm, detector = EDT) was used to detect secondary electrons onto dry samples operating under vacuum conditions at 61.90 KX (Azo-tz-PEDOT:PSS), 59.79 KX (N$_3$-PEDOT:PSS) and 57.64 KX (PEDOT:PSS) of magnifications.

### Electrochemical and electrical characterisation

**Cyclic voltammetry.** CV analyses were carried out with Autolab PGSTAT302N (Metrohm Italiana S.r.l, Italy) potentiostat/galvanostat interfaced with a personal computer, equipped with NOVA software. The films characterisation was performed with an ITO-coated square glass slide (25 mm × 25 mm, Kintec, Hong Kong) as WE, (Ag/AgCl NaCl (3 M) (Redox.me, Sweden) as RE, a platinum wire (Merck Life Science S.r.l., Italy) as CE and PBS solution (Merck Life Science S.r.l., Italy), with pH = 7.4 (100 μL) as supporting electrolyte and applying a scanning voltage from −0.4 V to +1.0 V at a scan rate of 50 mV s$^{-1}$.

**Electrochemical impedance spectroscopy.** EIS analyses were carried out with Autolab PGSTAT302N (Metrohm Italiana S.r.l, Italy) potentiostat/galvanostat interfaced with a personal computer, equipped with NOVA software. The films characterisation was performed with an ITO-coated square glass slide as WE, Ag/AgCl NaCl (3 M) (Redox.me, Sweden) as RE and CE and a PBS solution (Merck Life Science S.r.l., Italy) with pH = 7.4 (50 μL). Impedance was measured at +0.01 V and −0.01 V as biased potentials in the frequency range from 10$^5$ Hz to 0.1 Hz.

### OPECT operation

Following characterisations were performed by using a commercial platform (ARKEO, Cicci Research, Italy).

All measurements have been carried out by using PBS (Merck Life Science S.r.l., Italy; pH = 7.4) as electrolyte solution.

**Characterisation of transistor devices: transfer-characteristic and output curves under dark and UV light conditions.** Devices were characterised by using two measurement probes connected to the source and drain electrodes and a third one connected to the azo-tz-PEDOT:PSS thin film employed as planar gate electrode. Transfer curves were taken by sweeping the gate voltage from −0.2 V to 0.8 V with a scan rate of 50 mV s$^{-1}$ and the drain voltage (both versus the source potential) from −0.8 V to 0.1 V with a scan rate of 50 mV s$^{-1}$. By using the same setup and parameters, devices were exposed with a UV lamp ($\lambda$ = 365 nm) at a power of 0.56 W/cm$^2$, 1.7 W/cm$^2$ and 2.8 W/cm$^2$ for 2 min and then the transfer-characteristic and output curves were acquired keeping turned the lamp on during the measurement.

**Characterisation of transistor devices: pulsed measurements under UV light illumination.** Pulsed measurements on devices were performed by using the previous platform and setup, keeping a fixed drain voltage of $V_{ds}$ = −200 mV and by applying different voltage square pulses at the gate electrode (PW = 500 ms, 1 s and 5 s and $V_{gs}$ = +300 mV) fixing the power of UV light which was set at 0.56 W/cm$^2$, 1.7 W/cm$^2$ and 2.8 W/cm$^2$ for each measurement.

**Characterisation of transistor devices: light pulsed measurements for charge calculation.** The measurements were performed by using the previous setup, keeping fixed drain and gate voltages at $V_{ds}$ = −200 mV and $V_{gs}$ = 0 mV, respectively. The gate electrode was exposed to an alternance of 60 s in dark and 60 s of UV light ($\lambda$ = 365 nm, 0.56 W/cm$^2$, 1.7 W/cm$^2$ and 2.8 W/cm$^2$ of light intensity).

**Characterisation of neuromorphic behaviour: light pulsed measurements.** Neuromorphic behaviour of devices was characterised by connecting two measurement probes to the source and drain electrodes and by connecting a third one to the azo-tz-PEDOT:PSS gate electrode, keeping a fixed drain and gate voltage of $V_{ds}$ = −200 mV and $V_{gs}$ = 0 mV, respectively and by irradiating the gate electrode with 500 light pulses ($\lambda$ = 365 nm, 2.8 W/cm$^2$ of light intensity, PW = 1 s and $\Delta t$ = 1 s). After the illumination of the sample, the device was kept at $V_{gs}$ = 0 mV in dark for 30 min.

### OFF-pathway simulation of retina

**Operation of retina circuit.** The resistor and the azo-OPECT were connected in series to each other by using a tin-based soldering and silver paint; the measurements were performed using a multichannel setup of a commercial platform (ARKEO, Cicci Research, Italy) and PBS (Merck Life Science S.r.l., Italy; pH = 7.4) as electrolyte solution. A pulsed voltage supply $V_{DD}$ = +400 mV/0 mV was applied during the entire duration of the measurements. The potential of the azo-tz-PEDOT:PSS gate terminal was held at 0 mV for the first part of the measurements and then increased up to 300 mV and kept constant for 60 s, in dark conditions. Then, the same experiment was performed exposing the device to UV light ($\lambda$ = 365 nm, 2.8 W/cm$^2$ of light intensity).

### Optoelectronic memory

**Characterisation of optoelectronic memory: writable and erasable behaviour.** All measurements were carried out by using PBS (Merck Life Science S.r.l., Italy; pH = 7.4) as electrolyte solution.

(1) The optoelectronic memory was characterised by connecting two measurement probes to the source and drain electrodes and by connecting a third one to the azo-tz-PEDOT:PSS gate electrode, keeping a fixed drain and gate voltage of $V_{ds}$ = −200 mV and $V_{gs}$ = 0 mV, respectively, and by irradiating the gate electrode with 500 light pulses ($\lambda$ = 365 nm, 2.8 W/cm$^2$ of light intensity, PW = 1 s and $\Delta t$ = 1 s). After the illumination of the sample, a negative voltage of $V_{gs}$ = −300 mV was applied at the gate terminal for 10 min. Finally, the potential at the gate was brought back to 0 mV for 2 min.

(2) The optoelectronic memory was characterised using the previous setup, keeping a fixed drain and gate voltage of $V_{ds}$ = −200 mV and $V_{gs}$ = 0 mV, respectively and irradiating the gate electrode with light pulses ($\lambda$ = 365 nm, 2.8 W/cm$^2$ of light intensity, PW = 2 s, and $\Delta t$ = 10 s). After the illumination of the sample, negative voltage pulses of PW = 2 s, $\Delta t$ = 10 s and amplitude $V_{gs}$ = −300 mV were applied at the gate terminal.

**Optoelectronic mechanism.** Two measurement probes were connected to the source and drain electrodes and a third one to the azo-tz-PEDOT:PSS gate electrode. The drain voltage was kept at $V_{ds}$ = −200 mV and negative voltage pulses of PW = 60 s and amplitude between $V_{gs}$ = −50 mV to $V_{gs}$ = −350 mV (with a step of −50 mV) were applied at the gate terminal.

### Atkinson-Shiffrin model

**Characterisation of learning by Atkinson-Shiffrin model.** The measurement was performed by connecting two measurement probes to the source and drain electrodes and by connecting a third one to the azo-tz-PEDOT:PSS gate electrode, and by using PBS (Merck Life Science S.r.l., Italy; pH = 7.4) as electrolyte solution. First, two voltage

square pulses were applied at the gate terminal ($V_{gs}$ = +200 mV, PW = 3 s, and $\Delta t$ = 15 s) in dark conditions, emulating the sensory stimulus stored in the sensory memory. Then the device was irradiated by UV light at low intensity (0.56 W/cm$^2$) and the sensory stimulus (two voltage square pulses) was delivered a second time. A memory erasure of 10 negative voltage square pulses ($V_{gs}$ = −300 mV, PW = 5 s, and $\Delta t$ = 12 s) was applied at the gate terminal. Lastly, the azo-OPECT was irradiated by UV light at high intensity (2.80 W/cm$^2$). The sensory stimulus and the memory erasure are performed once again.

## Computational details

The photophysical properties of azo-tz-PEDOT were investigated via state-of-the-art density functional theory (DFT) calculations with the Gaussian16 suite of Quantum Chemistry programs[41]. For all minimum-energy structural optimisation and ground state electronic properties, we applied a standard level of theory with the reliable B3LYP hybrid density functional[42] and Pople's double-z basis sets 6−31 G(d,p) for all atoms[43,44]. Vertical electronic excitation and excited state properties of cis and trans azo-tz-PEDOT systems were characterised with time-dependent DFT (TD-DFT) calculations at the same level of theory[45].

## Data analysis

Numerical data analysis was carried out through custom-made MATLAB and python scripts.

Current (conductance) modulations were computed through a MATLAB script. In particular, the script allows to select two data points of current raw data (typically starting and ending points) and compute the difference between the two. Subsequently, such difference was converted into a percentage variation, using the value of the first data point (starting point of the raw current data) as reference, i.e., 100%. Statistical analysis, i.e., average, and standard deviation, was performed by exploiting MATLAB built-in functions.

Numerical fittings of EIS data were performed through a custom-made python script. The fitting procedure started by manually defining the most suited circuital model to fit the data. Then, exploiting the least square optimisation function (NumPy package), the parameters of the chosen electrical circuits were adjusted to fit EIS measured data.

# Data availability

The raw data and codes generated in this study are available upon request to the corresponding authors.

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

## Acknowledgements

F.S. and V.C. acknowledge the support of the European Research Council starting Grant BRAIN-ACT No. 949478. The authors acknowledge Dr. Alessandro Emendato for the experimental support on the uclear magnetic resonance spectroscopy and Dr. Lorenzo Zani, Dr. Viviana Rincón Montes and Prof. Alberto Salleo for the scientific discussion corroborating the work. The authors also thank Dr. Ziyu Gao, Dr. Alessandro Ruffoni and Prof. Daniele Leonori for support on electro-chemical experiments.

## Author contributions

F.C. and U.B. contributed equally to the experimental design, data acquisition, analysis. M. Prato, A.C., V.C., S.F. and C.C. contributed to data acquisition (material characterisation). A.M., A. B.M.G. and M. Pavone contributed to data acquisition and analysis (simulations). O.B. contributed to experimental design, data acquisition, analysis, and supervision. F.S. contributed to conceiving, funding acquisition, supervision and data analysis. All authors contributed to manuscript writing and revision.

## Funding

## Competing interests

The authors declare no competing interests.
