## [Peer Review File · Nature Communications]

REVIEWER COMMENTS

Reviewer #1 (Remarks to the Author):

The authors had synthesized azobenzene-functionalized PEDOT:PSS as the gate electrode and assembled it with PEDOT:PSS channel to construct a multimodal OPECT platform for neuromorphic units. Various methods were utilized to characterize the synthetic gate materials. Taking advantages of the photo-electrical properties of the functionalized gate, the authors had showed the first example of OPECT-based neuro-mimicked device. I recommend the publication of this manuscript if the following issues are properly addressed.

Major issue:

1. P4, line 104 - 106, "cyclic voltammetry measurements showed two characteristic peaks that can be attributed to the reduction of the azobenzene N=N bond induced by the presence of acidic protons in the electrolyte solution." The explanation on the cause of the oxidation current was missed. Was the reaction a reversible one? Moreover, the pH of the electrolyte solution should be noted.
2. P5, line 124 - 129, the definition of HOMO, HOMO-x, LUMO and LUMO+x that linked to the Figure 1g were confused. "Here, the levels alignment is not affected by the conformational orientation," this claim conflicted with the difference in the UV-Vis absorption spectra of two isomers. "Suggesting that the cis/trans isomerization is not a key process in the charge transfer mechanism," what was the key process in the present case, what role did the isomerization play in the whole process, and was the claim contradictory with the CV results that suggesting the isomerization would create additional impediment?
3. In Figure 3f, the authors applied 500 light pulses to modulate the channel and the result of synaptic recovery demonstrated the LTP. The signal of electrical stimulation should also be shown, what about its LTP?
4. Figure 4b-ii and Figure S12.3, compared with the saturated modulation towards the channel using azo-tz-PEDOT:PSS at gate voltage of -120 mV, why the PEDOT:PSS gate showed no saturated modulation even at -300 mV? Between the light on and off, why the PEDOT:PSS gate showed different modulation behaviors at positive gate voltage, and showed similar behaviors at negative gate voltage?
5. In the transient channel current measurements, the baselines before a light or electrical pulse were different in many figures. The authors should carefully explain the impact of the baseline level on the gain or response of the channel current when a stimulus was exerted.
6. The corresponding photo-induced signal mechanism should be explained clearly. According the UV visible spectra in Figure 1c, the absorption was affected by the conformational orientation and thus the charge transfer capability.

7. Different conformational orientation could induce different gate states and thus different channel current. But photo-induced charge transfer could induce gate Faraday current, and then channel current was modulated. These should be discussed.

Minor issue:

1. Figure S6, first 5 lines. The authors should indicate the semi-circular shape of PEDOT:PSS at high frequencies and of azo-tz-PEDOT:PSS at low frequencies in the figures.

2. Figure 1f, why the output curves of the device did not pass the zero point?

3. P7, line 155-156, "showing a photoexcitation process leading to the gate electrode polarization (positive charges accumulation)", if the positive charges accumulated at the gate, the channel current should increase.

4. P7, line 170, Figure 2e should be 2f. And the "e" in the figure was partially covered.

5. In Figure 2g, when VL was applied at 0 mV, VGC displayed multiple pulse steps, what causes the pulse signal?

6. The label in Figure 2g-ii was incorrect. "VL" should be modified as "VGC"

7. Figure S10.3, parallel experiments should be conducted.

8. P12, line 293 and Figure 4a, only one light pulse could be seen.

9. The information of S12 could not prove that the variation of the current values depends on the amount of protons. Because some other cations are also existed in PBS.

10. The data points of different pulse duration were so few that the trend was not obvious in Figure S10.3.

Reviewer #2 (Remarks to the Author):

This work synthesized a photo-switchable PEDOT:PSS that was integrated in an OPECT. The device exhibited current modulation upon different light stimulation conditions. More importantly, optoelectronic behaviour of this neurohybrid building block allowed to fabricate an "all-in-one" device, which was capable of both biomimic vertebrate retina visual pathways and to operate as an optical neuromorphic synaptic device capable to emulate both STP and LTP. This work demonstrates the potential of organic electrochemical transistors for photoelectric synapses, which has important implications for the development of implantable neuromorphic electronics. Therefore, I think this

work could be published in Nature Communications with major revisions. Some suggestions are as follows:

1. What is the charge mobility of OPECT.
2. What effect does the channel size (Channel length and channel width) of OPECTs have on its synaptic properties.
3. The response and recovery speed of the device to light are relatively slow. How does it correspond to the fast image imaging and switching of the retina.
4. When the light is off, what causes the current overcharge shown in Figure 4.
5. What is the power consumption of the device and a detailed comparison list of currently reported photoelectric synaptic devices should be given.

Reviewer #3 (Remarks to the Author):

A paper titled "Azobenzene-based organic opto-electronic transistor for neurohybrid building blocks" describes synthesis, fabrication and characterization of a light-sensitive PEDOT:PSS based OECT, termed organic photoelectrochemical transistor, or OPECT. The authors describe various aspects of device operation, including its use as a memory and (currently very popular) "neuromorphic" functioning (change in conductance).

In the opinion of the reviewer, the results appear novel and interesting. However, the overall device operation does not seem very impressive (e.g. the change in conductance is often noticeable, but very small). Also, there is a complete lack of information as to how reproducible (how many devices were characterized) and stable (two and a half periods shown in Fig 2c) the results are. As such, it is difficult to determine if the manuscript warrants publication in Nature Communications. In addition, authors are recommended to proof-read their manuscript (e.g. Introduction, Page 3, line 55 "...multimodal platforms capable to mimic...", or in Abstract, line 35, "... capable of interface properly...").

The image quality is really, really poor, making it very difficult, or sometimes impossible, to see any meaningful details (axis labels, numbers, etc.).

Below are specific comments.

Page 2, line 35. (also page 3, line 52) What's the base of saying that OPECT is more cost efficient? compared to what? – speculative. ITO (used by authors is also “expensive”, and gold, compared to some organic semiconductors, is not that expensive)

Page 3, line 54. Gold is actually pretty biocompatible, with very low toxicity.

Page 3, line 75. “The unprecedented...” – a bombastic claim

Page 6, line 134, Figure 1 caption. what material were the electrodes made from, ITO? this should be explicitly stated. what's the channel length? also, the length between S/D and gate?

Page 7, line 151. how reversible and stable is this? based on Fig2c reversibility and stability is questionable, especially over long term

Page 7, line 162. what's your definition of "almost instantaneous"? from Fig 2c, the horizontal axis is hundreds of seconds, so it's impossible to tell how "instantaneous" this change is (this should be clearly shown, either as separate subfigure or an inset), because in the currently shown figure it looks like there might be a slope, indicating that it is not instantaneous.

Page 7, line 164. what about after the first three pulses?

Page 7, line 169. what do you mean by "more complex electrical circuit"? that would indicate that you have previously shown a circuit, which you didn't (a single device is not a circuit). Also, this is an inverter, and you should / could refer to it as such. and perhaps you should then characterize it, in a form of a typical input/output curve.

Page 7, line 180. “... three retina cell...”. I don't understand what you mean by "three retina cell", i only see a single cell, "emulated" with a single OPECT device

Page 8, line 184. the figure quality is really low so it's very hard to see. Either a better quality image, or better explanation is needed.

Page 9, line 206, caption of Figure2c. That's only about 30% change. Also, in the figure the ON and OFF currents vary significantly with time. the IG ON seems to decrease with time, pretty significantly. only two and a half periods are shown? what does it look like after 20 or 100 switches/periods? also, what if period is extended in excess of 60 seconds. will the ON current decrease to the level of OFF current? 400 seconds seems very small to determine how "practical" this device is.

Page 9, line 219, caption of Figure2g. In about 300 seconds V_{CG} decreased from about 85 mV to about 60 mV. that's not very stable.

Page 10, line 224. "... an ideal neuromorphic platform...". what do you mean "ideal", under what conditions? ideal for what?

Page 10, line 226. "... when square voltage pulses...". but this is just a "conventional", shown many, many times before

Page 10, line 233. "... a dramatic decrease...". 12.5% change is noticeable but not dramatic.

Page 10, line 241. results presented in table S10 are a bit "troublesome". light intensity of 20% (with PW of 500 ms), results in 4.2% change. 60% light intensity results in 12.5% change. but 100% results in 9.5% change. those are not monotonic. The same is true when increasing the pulse width: at 20% light intensity, the change is 4.2% -> 2.7% -> 4.5%. In addition, examination of Figure S10.2 (which is really difficult because of extremely poor image/figure quality) reveals for each of the 9 graphs, the "base" channel current varies from about 200 uA to 400 uA (pretty spectacularly big device variability). that's a really big difference, of about 100%, which dwarfs the 4.2% change due to the application of light.

Page 10, line 253-267. Examination of Figure S11 makes it difficult to come up with any sort of conclusion. in (a) it indeed appears that the magnitude of the 2nd spike is a bit higher than the 1st one (this should be quantified, perhaps in a table), but is very small compared to the overall magnitude. what would happen if 3, 5, or 10 pulses were applied? In (b), top (with light) shows an overall drift of channel current, compared with bottom (without light). the apparent increase of current at the 2nd spike is present with and without light. it's difficult (or impossible) to draw any sorts of conclusions here. Also, again, there is no discussion as to how reproducible this effect is. was this observed in a single device or across multiple (how many?) devices? how much was the device-to-device variation.

Page 11, line 270, caption of Figure 3a. it is not clear how the experiment was actually design (how was the light positioned, what was its angle, was the gate electrode obstructing the light, etc.). a picture would be very instructive.

Page 13, line 322. what would happen if negative pulses were applied in the presence of light?

Reviewers' Comments to the manuscript

Reviewer

#1:

The authors had synthesized azobenzene-functionalized PEDOT:PSS as the gate electrode and assembled it with PEDOT:PSS channel to construct a multimodal OPECT platform for neuromorphic units. Various methods were utilized to characterize the synthetic gate materials. Taking advantage of the photoelectrical properties of the functionalized gate, the authors showed the first example of an OPECT-based neuromimicked device. I recommend the publication of this manuscript if the following issues are properly addressed.

Reviewer #1: Major issues

- 1 *P4, line 104 - 106, "cyclic voltammetry measurements showed two characteristic peaks that can be attributed to the reduction of the azobenzene N=N bond induced by the presence of acidic protons in the electrolyte solution." The explanation on the cause of the oxidation current was missed. Was the reaction a reversible one? Moreover, the pH of the electrolyte solution should be noted.*

We thank the reviewer for this comment. In the cyclic voltammetry measurements, the pH of the electrolyte (phosphate buffered saline -PBS) is 7.4. It contains species such as KH_2PO_4 able to protonate the "azo" bond (N=N) in the azobenzenes attached on the surface of the polymer films [1].

The cyclic voltammetry measurements also showed a cathodic peak at +0.57 V which could be correlated to the oxidation of the azobenzenes [2].

Since the difference between the anodic and cathodic peak potentials, called peak-to-peak separation (ΔE_p) = 0.69 V is not possible to assume that the reaction is reversible [3].

For clarity, we have refined the text in the manuscript and the pH of the PBS electrolyte was included in the **Experimental Section**.

Changes in the manuscript:

- Revision of text on **P4, Line 103 – 106**: 'In addition, cyclic voltammetry measurements showed two characteristic peaks: a reduction peak at –0.12 V that might be attributed to the reduction of the azobenzene N=N bond induced by the presence of protons from H_2PO_4 in the electrolyte solution and an oxidation peak at +0.57 V which might be due to the oxidation of the azobenzenes^{25,25}.
- Inclusion of the reference in the text **P4, Line 106**: Goulet-Hanssens, A. *et al.* Hole Catalysis as a General Mechanism for Efficient and Wavelength-Independent Z → E Azobenzene Isomerization. Chem 4, 1740–1755 (2018).
- Revision of text in **Experimental Section** on **P19, Line 515**: The films characterization was performed with an ITO-coated square glass slide (25 mm x 25 mm, Kintec, Hong Kong) as WE, (Ag/AgCl NaCl (3 M) (Redox.me, Sweden) as RE, a platinum wire (Merck Life Science S.r.l., Italy) as CE and PBS solution with pH = 7.4 (100 μL) as supporting electrolyte and applying a voltage from –0.4 V to +1.0 V at a scan rate of 50 mV s^{-1} .
- Revision of text in **Experimental Section** on **P20, Line 522**: 'The films characterization was performed with an ITO-coated square glass slide as WE, Ag/AgCl NaCl (3 M) (Redox.me, Sweden) as RE and CE and a PBS solution with pH = 7.4 (50 μL). Impedance was measured at +0.01 V and –0.01 V as biased potentials from 105 Hz to 0.1 Hz.'
- Revision of text in **Experimental Section** on **P20, Line 528 and P21, Line 567**: 'All measurements have been carried out by using PBS (pH = 7.4) as electrolyte solution.'
- Revision of text in **Experimental Section** on **P20, Line 558**: 'The resistor and the azo-OECT have been connected in series to each other by using a tin-based soldering and silver paint; the measurements have been performed using a multichannel setup of a commercial platform (ARKEO, Cicci Research) and PBS (pH = 7.4) as electrolyte solution.'
- Revision of text in **Experimental Section** on **P21, Line 588**: 'The measurement has been performed by connecting two measurement probes to the source and drain electrodes and by connecting a third one to the azo-tz-PEDOT:PSS gate electrode, and by using PBS (pH = 7.4) as electrolyte solution.'

- 2 *P5, line 124 - 129, the definition of HOMO, HOMO–x, LUMO and LUMO+x that linked to Figure 1g were confused. "Here, the levels alignment is not affected by the conformational orientation," this claim*

conflicted with the difference in the UV–Vis absorption spectra of two isomers. “Suggesting that the cis/trans isomerization is not a key process in the charge transfer mechanism,” what was the key process in the present case, what role did the isomerization play in the whole process, and was the claim contradictory with the CV results that suggesting the isomerization would create additional impediment?

We thank the reviewer for this comment. The statement has no contradiction regarding the energy level alignment and the UV–Vis spectra obtained for the cis and trans isomers. Indeed, the energy level alignment regards the relative positions of MO localized on different molecular moieties without any assumption on the energy gap between states involved in UV–Vis electronic transition. The energy gap has been found with TD-DFT calculations showing differences between cis and trans isomers, in agreement with the UV spectra.

In Figure 1g, the energy levels of both cis and trans isomers of the azo-tz-PEDOT are reported with the corresponding energetic values (eV). The HOMO-2 and LUMO, displayed on the left side of the energy diagram, represent the highest occupied and lowest unoccupied molecular orbital localized on the azo-tz moiety. On the right side, the HOMO and LUMO₊₁ refer to the highest occupied and lowest unoccupied molecular orbital localized on the PEDOT backbone.

For what concerns the discussion on the conformational orientation and the related implications on the charge transfer mechanism, a direct comparison of the cis/trans energy diagrams can be considered.

In fact, an additional figure has been included in the revised Supporting Information (Fig. S8.4) where the levels alignment is clearly retained in both conformers, with HOMO and LUMO levels being localized on the PEDOT backbone and on the azo-tz moiety, respectively.

Isomerization is not expected to be key in the whole charge transfer process, as the conformational orientation of azobenzene does not significantly alter the underlying relative energetics of azo and PEDOT molecular orbitals. On the other hand, the key process is driven by the photoinduced azo-to-PEDOT charge transfer/trapping mechanism. Upon UV light irradiation, an intramolecular charge transfer from the photoexcited azobenzene (LUMO) to the PEDOT backbone (HOMO) could take place. In principle, the formation of such spatially separated charge state can, result in charge trapping on the azo side, thus explaining the polarization at the gate electrode and the resulting decrease in the current flow.

Changes in the manuscript:

- Replacement of HOMO-x and LUMO+x labeling with the specific HOMO₋₂ and LUMO₊₁ in the revised version of the manuscript (**P5, Line 128-129**).
- Revision of caption in Figure 1g (**P7, Line 148-151**) to make clearer the results obtained from DFT/TDDFT calculations:

‘Chemical structure of the azo-tz-PEDOT system and energy levels of cis conformer obtained from DFT/TDDFT calculations. Only HOMO -2/LUMO levels localized on azo-tz (pink box) and HOMO/LUMO +1 localized on PEDOT (blue box) are displayed. The arrow shows the possible photoinduced electron transfer from the LUMO level (localized on the photoexcited azobenzene) to the HOMO level of the system (localized on the PEDOT backbone).’

- Revision of text on **P5, line 122-134**:

‘DFT/TD-DFT calculations were performed on a azo-tz-PEDOT model system to support the interpretation of the light-responsive mechanisms and to provide an unbiased energetic picture of the possible light-induced charge transfer processes (see Section S8 in SI for method validation). Our results confirmed the nature of the UV–vis transitions for cis and trans conformers as found in experiments (see Fig. S8.3 and Fig. 1c for comparison). For the cis conformer, we predicted the following energy diagram: the HOMO is localized on the PEDOT backbone, the LUMO is instead localized on the azo-tz moiety; the other relevant MO is the HOMO-2 that is localized onto the azo-tz moiety (see Fig. 1g). Within this energy framework, upon UV light irradiation, the photoexcited electron can be transferred from the LUMO (on the azo-tz) to the HOMO on the PEDOT backbone conductive system. Population of states located on the azobenzenes would lead to spatially separated charge carriers that, in principle, can be trapped on the azo side (positive charge highlighted in Fig. 1g). This mechanism would clarify the polarization at the gate electrode and the subsequent decrease in the current flow.’

- Revision of **Supporting Information S8**:

As shown in Fig. S8.4, the energy levels alignment obtained for the cis and trans conformers of azo-tz-PEDOT systems is similar, suggesting that the conformational orientation of azobenzene does not alter the

underlying energetics and thus the trans-cis isomerization is not expected to be a key process in the whole charge transfer process.

- Refinement of **Figure 1g**:

- Inclusion of Figure S8.4 in **Supporting Information S8.4**

Figure S8.4 Energy levels of trans (left) and cis (right) azo-tz-PEDOT obtained from DFT-TDDFT calculations: HOMO/LUMO orbitals from azobenzene and PEDOT moieties are highlighted in pink and blue, respectively. The values are reported in electronvolts. Corresponding density surfaces as plotted on the minimum-energy structures are also displayed to the side (isosurface level 0.01 eV/Å³).

- In Figure 3f, the authors applied 500 light pulses to modulate the channel and the result of synaptic recovery demonstrated the LTP. The signal of electrical stimulation should also be shown, what about its LTP?*

We thank the reviewer for the comment. The application of 300 pulses (PW = 1 s, Δt = 1 s and A = +300 mV) as gate bias results in conductance difference in terms of conditioning of ΔG% = 1.91 ± 1.70 and an extinction of ΔG% = -0.82 ± 1.25 with a recovery of 43% after 10 minutes (N = 3). In addition, when 500 light pulses have been applied, the conductance variation is ΔG% = 8.83 ± 0.42 for the conditioning and ΔG% = -2.19 ± 0.18 for extinction with a recovery of 25% after 30 minutes (N = 3) as reported in the figure below.

In conclusion, a lower "electrical" conditioning is achieved in comparison to light (pulses). Furthermore, the extinction is faster by applying only "electrical" pulses exhibiting a recovery of 43% instead of 25% achieved with the application of the light pulses, even after 10 minutes and resulting in a less effective LTP.

- 4 *Figure 4b-ii and Figure S12.3, compared with the saturated modulation toward the channel using azo-tz-PEDOT:PSS at gate voltage of -120 mV, why the PEDOT:PSS gate showed no saturated modulation even at -300 mV? Between the light on and off, why the PEDOT:PSS gate showed different modulation behaviors at positive gate voltage, and showed similar behaviors at negative gate voltage?*

We thank the reviewer for the comment. The saturation of the gate modulation was not observed, (given the geometry of the fabricated OECTs) as also shown in the following transcharacteristic at gate voltages between 0 and 750 mV

Furthermore, the difference between the PEDOT:PSS modulation under light and dark conditions is related to the mechanism proposed in Figure 4c (azobenzene reduction). Here, the negative voltage applied at the

gate terminal and the subsequent reduction of the azobenzene cause the attraction of protons from the electrolyte to the gate, competing with the charge injection induced by the light.

As a result, for a high (negative) gate bias, the light-induced effect is completely limited, achieving a modulation both under light and dark conditions. On the other hand, when a positive gate bias is applied, the light-induced channel current modulation is significantly different from the dark condition.

- 5 *In the transient channel current measurements, the baselines before a light or electrical pulse were different in many figures. The authors should carefully explain the impact of the baseline level on the gain or response of the channel current when a stimulus was exerted.*

We thank the reviewer for the comment. Considering the (statistical) sample size (at least 3 independent experiments) and the typical OECT channel drifts (i.e., due to swelling, device geometry), we performed statistical analysis on the channel current (conductance) variation percentage.

To clarify this, a “Data Analysis” paragraph was added in the Experimental Section.

Changes in the manuscript:

- Inclusion of the new paragraph ‘**Data Analysis**’ on **P22, Line 607-610**: “Numerical data analysis was carried out through custom-made MATLAB scripts. All current (conductance) modulations were computed as a percentage with respect to each device baseline. Statistical analysis was performed on current (conductance) variations as percentages.”.

- 6 *The corresponding photoinduced signal mechanism should be explained clearly. According the UV visible spectra in Figure 1c, the absorption was affected by the conformational orientation and thus the charge transfer capability.*

We thank the reviewer for the suggestion on how to improve the discussion. UV-vis spectroscopy detects electrons excitation associated with optical transitions among energy levels that underlie specific selection rules, while charge transfer phenomena may involve different energy levels.

Cis and trans conformers show very different response to UV light irradiation, with a low-intensity absorption peak at ~450 nm for the former and a high-intensity peak at ~350 nm for the latter. As a matter of fact, the electrons excitation calculated by means of DFT/TDDFT are in good agreement with UV-Vis results (see Figure S8.3 and Figure 1c for comparison). For both conformers, these electronic transitions involve energy levels localized on the azobenzene moieties and represent the HOMO-2/LUMO and HOMO-6/LUMO levels within the whole azo-tz-PEDOT system. By looking at the energy diagram and the HOMO position, we can see that the LUMO-to-HOMO intramolecular charge transfer may take place due to the level alignment, and this is not altered by the conformational orientation. We included this discussion in the revised version of the manuscript (**P5, line 122-134**).

Changes in the manuscript:

- Revision of text on **P5, line 122-134**:

‘DFT/TD-DFT calculations were performed on an azo-tz-PEDOT model system to support the interpretation of the light-responsive mechanisms and to provide an unbiased energetic picture of the possible light-induced charge transfer processes (see Section S8 in SI for method validation). Our results confirmed the nature of the UV-vis transitions for cis and trans conformers as found in experiments (see Fig. S8.3 and Fig. 1c for comparison). For the cis conformer, we predicted the following energy diagram: the HOMO is localized on the PEDOT backbone, the LUMO is instead localized on the azo-tz moiety; the other relevant MO is the HOMO-2 that is localized onto the azo-tz moiety (see Fig. 1g). Within this energy framework, upon UV light irradiation, the photoexcited electron can be transferred from the LUMO (on the azo-tz) to the HOMO on the PEDOT backbone conductive system. Population of states located on the azobenzenes would lead to spatially separated charge carriers that, in principle, can be trapped on the azo side (positive charge highlighted in Fig. 1g). This mechanism would clarify the polarization at the gate electrode and the subsequent decrease in the current flow.’.

- Caption correction **P7, Line 148-151: Figure 1g):** ‘Chemical structure of the azo-tz-PEDOT system and energy levels of cis conformer obtained from DFT/TDDFT calculations. Only HOMO -2/LUMO levels localized on azo-tz (pink box) and HOMO/LUMO +1 localized on PEDOT (blue box) are displayed. The arrow shows the possible photoinduced electron transfer from the LUMO level (localized on the photoexcited azobenzene) to the HOMO level of the system (localized on the PEDOT backbone).’

- Revision of **Supporting Information S8:**

As shown in Fig. S8.4, the energy levels alignment obtained for the cis and trans conformers of azo-tz-PEDOT systems is similar, suggesting that the conformational orientation of azobenzene does not alter the underlying energetics and thus the trans-cis isomerization is not expected to be a key process in the whole charge transfer process.

- 7 *Different conformational orientation could induce different gate states and thus different channel current. However, photoinduced charge transfer could induce gate Faraday current, and then channel current was modulated. These should be discussed.*

We thank the reviewer for the comment. The current amplitude resulting from the photoinduced charge transfer (hundreds of nanoampere) (Figure 2c) is such that no secondary species would be affected/interacting or redox reactions occur at the interface between the gate electrode and the electrolyte solution, thus limiting the possibility of inducing a faradaic current at the gate electrode therefore not modulating the conductance (channel current).

Reviewer #1: Minor issues

- 1 *Figure S6, first 5 lines. The authors should indicate the semicircular shape of PEDOT:PSS at high frequencies and of azo-tz-PEDOT:PSS at low frequencies in the figures.*

We thank the reviewer for the comment.

Changes in the manuscript:

- Inclusion of new labels in Figure S6a and Figure S6b in Supporting Information S6 to identify the semicircular shape of PEDOT:PSS at high frequencies and of azo-tz-PEDOT:PSS at low frequencies.

- 2 *Figure 1f, why the output curves of the device did not pass the zero point?*

We thank the reviewer for pointing out this error.

Changes in the manuscript:

- The mismatch between the points of the x-axis and the y-axis has been fixed and Figure 1f has been updated.

- 3 *P7, line 155-156, “showing a photoexcitation process leading to the gate electrode polarization (positive charges accumulation)”, if the positive charges accumulated at the gate, the channel current should increase.*

We thank the reviewer the comment. As PEDOT:PSS-based OECTs operate in depletion mode, the gating process (positive bias at the gate terminal) causes the injection of cations from the electrolyte to the bulk of the polymeric channel inducing dedoping and a decrease of the channel current [4]. Here, similarly the gate electrode is polarized with positive charged when irradiated by UV light. This phenomenon results in an effective gating of the transistor, reducing the current that can flow inside the polymeric channel.

Changes in the manuscript:

- Revision of text (P7, line 159-162): “Such positive charge accumulation at the gate electrode, induced by the UV light irradiation, resulted in an effective gating of the transistor. As a consequence, positive charged ions can penetrate the bulk of the polymeric channel, causing the dedoping of the PEDOT:PSS and resulting decrease of I_{DS} ”.

4 *P7, line 170, Figure 2e should be 2f. In addition, the “e” in the figure was partially covered.*

Changes in the manuscript:

- Revision of letters in **Figure 2** which is also provided in high resolution.

5 *In Figure 2g, when VL was applied at 0 mV, VGC displayed multiple pulse steps, what causes the pulse signal?*

We thank the reviewer for the comment. The system is provided of a power supply VDD which is alternated too and is responsible of the pulsed signal shape in VGC.

6 *The label in Figure 2g-ii was incorrect. “VI” should be modified as “VGC”*

Changes in the manuscript:

- Revision of label in **Figure 2gii** which is also provided in high resolution.

7 *Figure S10.3, parallel experiments should be conducted.*

At the best of our knowledge, PEDOT:PSS photoexcitation or its photoelectric effect upon UV illumination ($\lambda = 365$ nm) has been not reported in literature [5]. Therefore, it is reasonable that illumination of the control samples (bare PEDOT:PSS) does not induce any detectable current change during the operation of the OECT. Moreover, any absorption at $\lambda = 365$ nm has been observed in the UV–visible spectrum of PEDOT:PSS and any spectral change was detected even after the illumination, as reported in Figure S5.1.

However, we carried out the operation of PEDOT:PSS-based OECT (black solid line, figure below) and azo-OECT (red solid line, figure below) applying 60 seconds in dark and 60 seconds of UV light by fixing $V_{gs} = 0$ and $V_{ds} = -200$ mV. Here, it is possible to see no appreciable changes of gate current during the illumination of the bare PEDOT sample.

Changes in the manuscript:

- Revision of text (**P5-6, Line 118-122**): Interestingly, under light exposure ($t = 5$ minutes, $\lambda = 365$ nm, 6 W), the I_{ds} current shifts to lower values suggesting a photoinduced charge transfer/charge trapping mechanism occurring between the azobenzenes and the PEDOT:PSS backbone (Figure 1g), unlike of PEDOT:PSS alone which typically does not show photothermoelectrical behavior²⁸.

- Inclusion of the reference in the text **P6, Line 122**: Lan, X. *et al.* p–n hybrid bulk heterojunction enables enhanced photothermoelectric performance with UV–Vis–NIR light. *Nanoscale* 14, 18003–18009 (2022).

8 ***P12, line 293 and Figure 4a, only one light pulse could be seen.***

We thank the reviewer for the comment. We reported a single pulse (Figure 4a) to better highlight the behavior of the device under the diverse light/electrical conditions. However, an overview on a larger number of pulses is reported in Supporting information S12 (Figure S12.1 and S12.3).

9 ***The information of S12 could not prove that the variation of the current values depends on the amount of protons. Because some other cations are also existed in PBS.***

We thank the reviewer for the comment. In PBS, Na⁺ and K⁺ cations are present, but their total molar concentration is comparable with that of the 100 mM NaCl solution we used in the experiments mentioned in Supporting Information S12. Moreover, considering that in PBS H₂PO₄ could provide protons, which amount is two orders of magnitude bigger than in the NaCl solution, this suggests that protons are responsible for this mechanism. A similar mechanism is also reported in literature [6].

Changes in the manuscript:

- Revision of the text (**P12-13, Line 314-318**): ‘In fact, when a different electrolyte (i.e., an aqueous solution 100 mM of NaCl), containing two orders of magnitude less protons compared to PBS, was used, the application of consecutive negative voltage pulses induced a very poor modulation of Ids confirming that in the variation of the current protons in the electrolyte are involved in the mechanism ³² (Supporting information S12).’

10 ***The data points of different pulse duration were so few that the trend was not obvious in Figure S10.3.***

We thank the reviewer for the comment. Following this, we carried out additional measurements with a pulse width (PW) fixed at 3 seconds and PW = 8 seconds (while varying the UV light power = 0.56 W/cm², 1.7 W/cm², 2.8 W/cm²). As shown in the figure and table below, the conductance variation is independent of the different PW.

Figure S10.3

Light intensity	$\Delta G\%$ conditioning (6 minutes)				
	PW: 500 ms	PW: 1 s	PW: 3s	PW: 5s	PW: 8s
100%	9.5 ± 2.1	9.1 ± 1.3	5.5 ± 1.0	12.5 ± 3.1	6.0 ± 3.5
60%	12.5 ± 3.1	11.8 ± 8.4	6.4 ± 2.3	7.9 ± 1.7	2.3 ± 1.0
20%	4.2 ± 0.6	2.7 ± 0.7	3.8 ± 1.9	4.5 ± 2.5	2.9 ± 1.6

Table S10.1

Light intensity	$\Delta G\%$ extinction (3 minutes)				
	PW: 500 ms	PW: 1 s	PW: 3 s	PW: 5s	PW: 8 s
100%	-0.8 ± 0.9	-0.9 ± 1.2	-0.6 ± 0.2	-2.7 ± 3.1	-0.2 ± 1.4
60%	-0.7 ± 1.1	-2.0 ± 1.3	-0.6 ± 0.2	-0.7 ± 0.6	0.2 ± 0.3
20%	0.9 ± 1.0	-0.1 ± 1.6	-0.1 ± 0.6	-0.2 ± 0.6	0.9 ± 1.0

Table S10.2

Changes in the manuscript:

- **Figure S10.3, Table S10.1 and S10.2** (Table S10 has been splitted in two tables) have also been updated in **Supporting Information S10**.
- Revision of the caption of **Figure S10.3**: ‘Conditioning calculated as percentage conductance variation ($\Delta G\%$) by changing the light intensity of the UV light (20% , 60%, 100%, N = 3 of each) and the PW applied to the gate electrode (500 milliseconds, 1 second, 3 seconds, 5 seconds and 8 seconds, N = 3 of each)’.
- Revision of the caption of **Table S10.1**: ‘Conditioning calculated as percentage conductance variation ($\Delta G\%$) by changing the light intensity of the UV light (20% , 60%, 100%, N = 3 of each) and the PW applied to the gate electrode (500 milliseconds, 1 second, 3 seconds, 5 seconds and 8 seconds, N = 3 of each).’
- Revision of the caption of **Table S10.2**: ‘Extinction calculated as percentage conductance variation ($\Delta G\%$) by changing the light intensity of the UV light (20% , 60%, 100%, N = 3 of each) and the PW applied to the gate electrode (500 milliseconds, 1 second, 3 seconds, 5 seconds and 8 seconds, N = 3 of each).’

This work synthesized a photoswitchable PEDOT:PSS that was integrated in an OPECT. The device exhibited current modulation under different light stimulation conditions. More importantly, the optoelectronic behavior of this neurohybrid building block allowed the fabrication of an “all-in-one” device that was capable of both biomimicking vertebrate retina visual pathways and operating as an optical neuromorphic synaptic device capable of emulating both STP and LTP. This work demonstrates the potential of organic electrochemical transistors for photoelectric synapses, which has important implications for the development of implantable neuromorphic electronics. Therefore, I think this work could be published in Nature Communications with major revisions. Some suggestions are as follows:

Reviewer #2: Major issues

1 *What is the charge mobility of OPECT*

We thank the reviewer for the comment. The OPECT features a standard spin coated PEDOT:PSS channel. In particular, two different charge mobilities can be estimated, as such material conduct both holes and ions. Hole mobility in PEDOT:PSS is usually in the range $1 - 10 \frac{cm^2}{Vs}$ [7–9], while ion mobility is the range $1 - 2 \times 10^{-3} \frac{cm^2}{Vs}$, based on the formulation of the PEDOT:PSS dispersion [10].

Moreover, for clarity, we have also modified the text to distinguish between ions migration and charge mobility which are two key aspects of our devices.

Changes in the manuscript:

- Revision of text on **P12, Line 298**: “To further investigate this effect on ions migration induced by the negative voltage bias at the azo-tz-PEDOT:PSS gate electrode, a series of consecutive pulses of increasing intensity (from -20 mV to -120 mV) were employed (Figure 4b)”.

2 *What effect does the channel size (Channel length and channel width) of OPECTs have on its synaptic properties.*

We thank the reviewer for the comment. Diverse channel sizes (length and width) have been not tested here due to the spatial resolution achievable (at most 4 x 7 mm as length x width) with our fabrication process.

However, OPECTs exploit the same working principle of OECTs, but with a gate bias modulated through light. In principle, the charge injection from the gate to the polymeric channel can be modulated decreasing the channel length (while keeping fixed the channel width) and thus increasing the gating efficiency of the transistor [4,11].

Here, given the device gating through UV light irradiation at the azo-tz-PEDOT:PSS gate electrode, the reduction of the channel length (maximization of the geometrical ratio) similarly would enhance the synaptic effect, as the whole gating efficiency would increase.

For clarity, we have given a perspective on this important (geometrical) aspect in the text.

Changes in the manuscript:

- Revision of text on **P15, Line 370-372**: “In perspective, by optimizing the geometrical ratio ($\frac{Wt}{L}$) of the OPECT channel, the light-mediated gating efficiency could be finely tuned, also reducing the switching speed of the device”.

3 *The response and recovery speed of the device to light are relatively slow. How does it correspond to the fast image imaging and switching of the retina.*

We thank the reviewer for the comment. Here, the main goal is to establish a proof of concept for the optoelectronic transduction mechanism and a retina-inspired synaptic behavior. The response time is strongly dependent on the large channel dimensions of our devices.

Indeed, the fast dynamics of the retina could potentially be achieved with the optimization of the geometrical ratio of the OPECT that governs the time response of the overall device (as also mentioned above). This aspect has been further discussed in the conclusion part and given as outlook.

Changes in the manuscript:

- Revision of text on **P15, Line 370-372**: “In perspective, by optimizing the geometrical ratio ($\frac{Wt}{L}$) of the OPECT channel, the light-mediated gating efficiency could be finely tuned, also reducing the switching speed of the device”.

4 ***When the light is off, what causes the current overcharge shown in Figure 4.***

We thank the reviewer for the comment. The decrease of current that can be observed in Figure 4 (dark condition) is a drift typical of PEDOT:PSS-based OECTs. This effect disappears after few seconds after the start of the measurement.

5 ***What is the power consumption of the device and a detailed comparison list of currently reported photoelectric synaptic devices should be given.***

We thank the reviewer for the comment. The power consumption of an OPECT can be assessed in different ways [12] For instance:

$$dE = S \times P \times dt \quad (1)$$

Where S is the area of the device, P is the power density of the optical stimulation and dt is the duration of the stimulation. In this way, the energy consumption is computed as the energy of the optical stimulus.

Another way to compute the energy consumption is:

$$dE = V \times I \times dt \quad (2)$$

In which V is the voltage of the device, I is the current of the device and dt is the duration of the optical stimulation. This formula relates the energy consumption of the device to the electrical response of the optoelectronic device.

Considering (2), the energy consumption of the presented device is $\sim 30 \mu J$. This value is several orders of magnitude higher than the power consumption of similar optoelectronics neuromorphic three-terminal devices, as shown in the following table. Such gap may be mainly ascribed to the overall area of the transistor channel ($L \cdot W$), that crucially hampers an energy efficient optoelectronic transduction [13]. In addition, as it is reasonable to assume that contact resistance plays a significant role in the energy consumption, the interface between the PEDOT:PSS channel and the ITO contact pads may further limit the energy efficiency.

Material	Area	Energy consumption	Ref
azo-PEDOT:PSS	$4000 \times 7000 \mu m^2$	$30 \mu J$	This work
Si-NM/MAPbI3	$25 \times 500 \mu m^2$	$1 pJ$	[14]
In2O3/ion-gel	$80 \times 1600 \mu m^2$	$100 pJ$	[15]
CsBi3I10/single-walled CNT	$30 \times 1000 \mu m^2$	$1 pJ$	[16]
PDVT-10/PVP + CsPbBr3 QDs	$30 \times 1000 \mu m^2$	$4.1 pJ$	[17]
IZO	$30 \times 100 \mu m^2$	$350 pJ$	[18]
SWCNT	$20 \times 1000 \mu m^2$	$2.5 nJ$	[19]
ITO/chitosan	$80 \times 1000 \mu m^2$	$390 \mu J$	[20]
IGZO/nanogranular SiO2	$80 \times 10 \mu m^2$	$1.1 pJ$	[21]
IGZO	$10 \times 100 \mu m^2$	$600 pJ$	[22]
IGZO/Al2O3	$20 \times 40 \mu m^2$	$12.5 pJ$	[23]

PEDOT/SnOx/IGZO	$500 \times 800 \mu\text{m}^2$	1 nW	[24]
InAs nanowire	$0.387 \times 6.2 \mu\text{m}^2$	10 nJ	[25]
pentacene/PMMA/CsPbBr ₃	$500 \times 1000 \mu\text{m}^2$	1.4 nJ	[26]
SiNCs	$10 \times 120 \mu\text{m}^2$	140 pJ	[27]
graphene oxide-IGZO/Ion-gel	$10 \times 10 \mu\text{m}^2$	150 pJ	[28]
single-walled CNT/graphene	$90 \times 30 \mu\text{m}^2$	150 nJ	[29]

Reviewer 3:

A paper titled “Azobenzene-based organic opto-electronic transistor for neurohybrid building blocks” describes the synthesis, fabrication and characterization of a light-sensitive PEDOT:PSS-based OECT, termed an organic photoelectrochemical transistor, or OPECT. The authors describe various aspects of device operation, including its use as a memory and (currently very popular) “neuromorphic” functioning (change in conductance).

In the opinion of the reviewer, the results appear novel and interesting. However, the overall device operation does not seem very impressive (e.g., the change in conductance is often noticeable, but very small). Additionally, there is a complete lack of information as to how reproducible (how many devices were characterized) and stable (two and a half periods shown in Fig 2c) the results are. As such, it is difficult to determine if the manuscript warrants publication in Nature Communications. In addition, authors are recommended to proof-read their manuscript (e.g. Introduction, Page 3, line 55 “...multimodal platforms capable to mimic...”, or in Abstract, line 35, “... capable of interface properly...”).

The image quality is truly, truly poor, making it very difficult, or sometimes impossible, to see any meaningful details (axis labels, numbers, etc).

Below are specific comments.

Reviewer #3: Major issues

- 1 *Page 2, line 35. (also page 3, line 52) What's the base of saying that OPECT is more cost efficient? compared to what? – speculative. ITO (used by authors is also “expensive”, and gold, compared to some organic semiconductors, is not that expensive)*

We thank the reviewer for the comment. Indeed, we have modified the abstract text to better highlight the key features of conjugated polymers being integrated into light-driven devices especially highlighting the potential of these materials to interface biological systems rather than focusing on cost-effective solutions.

Along the same line, the discussion on conventional metals and semiconductors was also refined on Page 3 (Introduction) with particular focus on the requirement for ion sensitivity as well as key features for cell interfacing.

Changes in the manuscript: revised Abstract.

- ‘The light–matter interplay to realize advanced light responsive multimodal platforms is an emerging strategy to engineer bioinspired systems such as optoelectronic synaptic devices. However, existing neuroinspired optoelectronic devices rely on complex processing of hybrid materials which not often exhibit required features for biological interfacing such as biocompatibility and low Young’s modulus. In this scenario, organic photoelectrochemical transistors (OPECTs) have recently paved the way toward a multimodal devices that can be better coupled to biological systems benefiting from conjugated polymers characteristics. In this scenario, neurohybrid OPECTs can be built to optimally interface neuronal systems as well as resemble typical plasticity-driven processes to potentially create more sophisticated integrated architectures between neuron and neuromorphic ends. Here, an innovative photoswitchable PEDOT:PSS was synthesized and successfully integrated in an OPECT. Such azobenzene-based organic neuro-hybrid building block was employed to mimic the retina structure exhibiting the capability to emulate visual pathways. Moreover, dually operating the device with opto and electrical functions, a light-dependent conditioning and extinction processes were achieved faithful mimicking synaptic neural functions such as short- and long-term plasticity.’
- Revision of text **P3, Line 53-59**: ‘However, traditional inorganic and hybrid optoelectronic platforms find major limitations in biologically driven systems where ion sensing as well as tailored biomechanical properties are required^{7,11}. Thereby, in order to fabricate multimodal platforms capable to mimic a potentially also directly interface biological systems such as neuronal cells, several strategies have been proposed for the development of optoelectronic devices featuring electromechanical stability in aqueous environment, ionic-electronic transduction and biophysical properties (i.e., biocompatibility, low stiffness¹³⁻¹⁵).’

2 **Page 3, line 54. Gold is actually pretty biocompatible, with very low toxicity.**

We thank the reviewer for the comment. As mentioned in the previous comment, we have revised the text in the introduction to better highlight the requirements of optoelectronic devices to both mimic and potentially interface biological systems other than biocompatibility.

Changes in the manuscript:

- Revision of text **P3, Line 53-59**: ‘However, traditional inorganic and hybrid optoelectronic platforms find major limitations in biologically driven systems where ion sensing as well as tailored biomechanical properties are required^{7,11}. Thereby, in order to fabricate multimodal platforms capable to mimic a potentially also directly interface biological systems such as neuronal cells, several strategies have been proposed for the development of optoelectronic devices featuring electromechanical stability in aqueous environment, ionic-electronic transduction and biophysical properties (i.e., biocompatibility, low stiffness¹³⁻¹⁵).’

3 **Page 3, line 75. “The unprecedented...” – a bombastic claim**

We thank the reviewer for the comment. We revised the introduction of the manuscript highlighting the advantage of having achieved the synthesis of a light-sensitive conjugate polymer considering a more appropriate vocabulary.

Changes in the text:

- Revision of text **P3, Line 73-77**: ‘Interestingly, this newly synthesized light-responsive and its integration conjugated polymer and its integration as gate electrode of this neurohybrid building block allowed to achieve and “all-in-one” device capable of both biomimic vertebrate retina visual pathways and to operate as an optical neuromorphic synaptic device capable to emulate both short and long term plasticity (STP and LTP).’ ...

4 **Page 6, line 134, Figure 1 caption. what material were the electrodes made from, ITO? this should be explicitly stated. what's the channel length? also, the length between S/D and gate?**

We thank the reviewer for the comment. We provided details on the substrate geometry, fabrication and illumination conditions giving the necessary revision of the text.

Changes in manuscript

- Revision of **Figure 1a** legend.
- Inclusion of ITO glass substrate schematics as **Figure S7.2** in **Supporting Information S7**.
- Revision of text in **Experimental Section** on **P17, Line 439**: ‘Organic electrochemical transistor fabrication. Customized patterned ITO coated glass (Surface resistivity $20 \Omega \text{ cm}^{-2}$, 25 mm x 25 mm, Kintec, Hong Kong), see also Figure S7.2, were cleaned in an ultrasonic bath with Alconox® detergent solution (Merck Life Science S.r.l., Italy), deionized water, acetone (Merck Life Science S.r.l., Italy), and 2-propanol (Merck Life Science S.r.l., Italy) (10 minutes for each solvent) and dried with compressed air.’
- Revision of text in **Experimental Section** on **P18, Line 455-460** to specifying gate and channel area as well as illumination area at the gate: ‘Finally, to remove the excess of PEDOT:PSS and pattern the OECT channel, a plasma etching (15 minutes, 100 W) was performed including a hard PDMS mask (15 mm x 7 mm, 23 mm x 12 mm) to cover both the planar gate and the channel with two PDMS masks. The so obtained OECTs have a channel area of 15 mm x 7 mm, a gate area of 13 mm x 10 mm of which the illuminated area is 20 mm². Before the use, the device was gently rinsed in deionized water overnight to promote the PEDOT:PSS swelling.’
- **Figure S7** has been renamed as **Figure S7.1** in **Supporting Information S7**.

Figure S7.2: Customized patterned ITO coated glass.

- 5 *Page 7, line 151. how reversible and stable is this? based on Fig 2c reversibility and stability is questionable, especially over long term*

We thank the reviewer for the comment. The stability of the light-induced current decrease was investigated in Figure 3a and Figure 3f. In the former, a continuous light stimulus was applied while monitoring the channel current. As a result, the change of channel current (conductance) is retained over a long period of time. In fact, more than three minutes after the removal of the light stimulation, the conductance level is no able to go back to the original level. In addition, in the latter, the continuous light stimulus is replaced by a series of light pulses. Here, the current (conductance) decrease is retained over a longer period after the stop of the light stimulation.

Furthermore, reversibility is shown in Figure 4a, where the application of a negative voltage bias was exploited to take the conductance back to (almost) the original level.

- 6 *Page 7, line 162. what's your definition of "almost instantaneous"? from Fig 2c, the horizontal axis is hundreds of seconds, so it is impossible to tell how "instantaneous" this change is (this should be clearly shown, either as separate subfigure or an inset), because in the currently shown figure it looks like there might be a slope, indicating that it is not instantaneous.*

We thank the reviewer for the comment.

Here, we highlighted the difference (as order of magnitude) of the trans-cis isomerization rate of the azobenzenes (minutes) compared with the rate of current change observed in the experiment (Figure 2cii).

This hypothesis was also confirmed by the UV spectra reported in Supporting Information S5 where the reversible cis-trans isomerization was studied. The recovery of the trans-isomer was possible after heating for 10 minutes ($T = 80^{\circ}\text{C}$) or irradiation with blue light for 5 minutes: here external stimuli facilitate the cis-trans isomerization occurring in the range of minutes and is probably even slower in absence of stimuli.

For this reason, we concluded that the charge transfer is mainly due to the photoinduced charge transfer in the Azo-PEDOT system, and the cis-trans isomerization contribution can only have a slight influence within the first light pulse, during which trans-cis isomerization occurs.

Conversely, for the following pulses, the reverse cis-trans isomerization does not have time to occur, so the change in the current can be justified by only by the photoinduced charge transfer in the Azo-PEDOT system.

We have clarified this point in the text **P7, Line 165-168**.

Change in the manuscript:

- Revision of text on **P7, Line 165-168**: 'Indeed, the spontaneous cis-trans relaxation occurs in the order of minutes in similar azobenzene monomers²⁵, while here the current decrease observed in the azo-OECT is in the order of seconds (10-30s), reaching the initial current values. This suggest that the cis-trans isomerization contribution can only have an influence within the first light pulse^{25,26}.'

7 *Page 7, line 164. what about after the first three pulses?*

We thank the reviewer for the comment. We performed additional measurements applying 20 light pulses, also reducing the pulse width from 60 to 2 seconds.

The time interval between pulses was 10 seconds so we could better evaluate the charge accumulation induced by switching the light on.

Here, the resulting charge at the gate electrode remains constant for each pulse, also after the first three pulses as shown in the figure below (charge calculation reported in Supporting Information S9).

8 *Page 7, line 169. what do you mean by "more complex electrical circuit"? that would indicate that you have previously shown a circuit, which you did not (a single device is not a circuit). Additionally, this is an inverter, and you should/could refer to it as such. and perhaps you should then characterize it, in a form of a typical input/output curve.*

We thank the reviewer for the comment and pointing out the misleading structure of that sentence in the manuscript.

In addition, as correctly pointed out by the reviewer, the circuit employed in this application has the same structure of a resistor-transistor logic (RTL) inverter. However, here the input is a light-dependent signal while the output is a voltage which would not resemble a traditional inverter.

We also did not tackle into this as the manuscript core is the validation of the (light driven) memory capabilities of the platform. Certainly, this could be further explored in future experiments.

Changes in the manuscript:

- Revision of text at **P7, Line 174**: “The azo-OECT was then integrated into an electrical circuit to implement a biohybrid retinal architecture.”

9 *Page 7, line 180. "... three retina cell...". I do not understand what you mean by "three retina cell", i only see a single cell, "emulated" with a single OPECT device.*

We thank the reviewer for the comment. The word ‘three’ has been deleted and the language revised. Furthermore, here, the azo-tz-PEDOT:PSS gate acts as the photoreceptors layer, the electrolyte-channel interface mimics the BCs layer, while the PEDOT:PSS based organic resistor acts as the GCs (Figure 2 e-f). In brief, the output voltage VGC is determined by the conductance of the azo-OECT channel which is controlled through the light input.

Changes in the manuscript:

- Revision of text **P8, Line 185**: ‘In the light of this, the retina cell layers’

- 10 *Page 8, line 184. the figure quality is truly low so it is very hard to see. Either a better quality image, or better explanation is needed.*

We thank the reviewer for the comment. All figures have been updated and saved in high resolution.

- 11 *Page 9, line 206, caption of Figure 2c. That is only approximately 30% change. Additionally, in the figure the ON and OFF currents vary significantly with time. the IG ON seems to decrease with time, pretty significantly. only two and a half periods are shown? what does it look like after 20 or 100 switches/periods? also, what if period is extended in excess of 60 seconds. will the ON current decrease to the level of OFF current? 400 seconds seems very small to determine how "practical" this device is.*

We thank the reviewer for the comment. Indeed, the reported gate current variation is generated by the application of the UV light alone and it is sufficient to modulate the channel current. In addition, the gate current can be further increased by applying simultaneously a positive voltage.

As also anticipated in the comment #7, the gate current decreases to the level of OFF current after the application of 20 light pulses with $PW = 2$ seconds and $\Delta t = 10$ seconds.

- 12 *Page 9, line 219, caption of Figure 2g. In approximately 300 seconds V_{CG} decreased from approximately 85 mV to approximately 60 mV. that is not very stable.*

We thank the reviewer for the comment. The measurement shown in Figure 2g is conducted under light condition. As a result, the voltage V_{CG} is supposed to decrease. In fact, the light irradiation changes the conductance of the transistor, resulting in a voltage decrease. As a comparison, the middle plot shows the same measurement under dark condition, in which the voltage is not decreasing (up to 300 s, while it then starts to decrease because of the application of a voltage input at the gate terminal).

- 13 *Page 10, line 224. "... an ideal neuromorphic platform...". what do you mean "ideal", under what conditions? ideal for what?*

- 14 *Page 10, line 226. "... when square voltage pulses...". but this is just a "conventional", shown many, many times before*

We thank the reviewer for both comments. The text has been revised to follow both suggestions.

Changes in the manuscript:

- Revision of text **P10, Line 228-231**: 'Indeed, as previously shown³⁵, platform can mimic neuromorphic features when square voltage pulses are applied at the gate terminal, acting as action potentials at presynaptic ends in biological neuronal synapses and modulating the conductance at the channel (post-synaptic end)³⁵ (Figure 3a).'

- Inclusion of the reference at **P10, Line 228**: Keene, S. T. A biohybrid synapse with neurotransmitter-mediated plasticity. Nature Materials 19, 16 (2020).

15 *Page 10, line 233. "... a dramatic decrease...". 12.5% change is noticeable but not dramatic.*
We thank the reviewer for the comment. The word 'dramatic' has been removed.

Changes in the manuscript:

- Revision of text **P10 Line 234-235**: 'decrease in the conductance of the postsynaptic channel was observed'...

16 *Page 10, line 241. results presented in table S10 are a bit "troublesome". light intensity of 20% (with PW of 500 ms), results in 4.2% change. 60% light intensity results in 12.5% change. but 100% results in 9.5% change. those are not monotonic. The same is true when increasing the pulse width: at 20% light intensity, the change is 4.2% -> 2.7% -> 4.5%. In addition, examination of Figure S10.2 (which is truly difficult because of extremely poor image/figure quality) reveals for each of the 9 graphs, the "base" channel current varies from approximately 200 uA to 400 uA (pretty spectacularly big device variability). that is a truly big difference, of approximately 100%, which dwarfs the 4.2% change due to the application of light.*

We thank the reviewer for the comment. The aforementioned nonmonotonic trend has been discussed in the main text on **P10, Lines 239-244**, highlighting also the uncorrelation between the electrical and optical stimulation, as the current decrease does not change as a function of the applied PW (Figure 3D and Figure S10.3).

Furthermore, statistical analysis on the current variation has been performed considering at least N=3 independent experiments.

To clarify this, a "Data Analysis" paragraph was added in the Experimental Section.

Changes in the manuscript:

- Inclusion of the new paragraph 'Data Analysis' on **P22, Line 607-610**: "Numerical data analysis was carried out through custom-made MATLAB scripts. All current (conductance) modulations were computed as a percentage with respect to each device baseline. Statistical analysis was performed on current (conductance) variations as percentages."

17 *Page 10, line 253-267. Examination of Figure S11 makes it difficult to come up with any sort of conclusion. in (a) it indeed appears that the magnitude of the 2nd spike is a bit higher than the 1st one (this should be quantified, perhaps in a table), but is very small compared to the overall magnitude. what would happen if 3, 5, or 10 pulses were applied? In (b), top (with light) shows an overall drift of channel current, compared with bottom (without light). the apparent increase of current at the 2nd spike is present with and without light. it is difficult (or impossible) to draw any sorts of conclusions here. Additionally, again, there is no discussion as to how reproducible this effect is. was this observed in a single device or across multiple (how many?) devices? how much was the device-to-device variation.*

We thank the reviewer for the comment. The PPF index was not evaluated for more than two consecutive pulses, following the physiological behavior of biological neurons that may exhibit potentiation or depression based on the time interval between two consecutive presynaptic stimuli [31]. All measurements were conducted on N=3 devices. The device-to-device variation was taken into account as discussed in the previous comment of the reviewer (#16).

Last, PPF indexes calculated under both light and dark conditions are comparable among all the samples, suggesting that this synaptic feature is not influenced by UV-irradiation.

For clarity, the text on Page 10-11, Line 262-264 was modified.

Changes in the manuscript:

- Revision of text on **P10-11, Line 262-264**: "Finally, PPF indexes calculated under dark and light conditions are comparable, suggesting that STP is independent of the light exposure and mainly induced by the electrical stimulation".
- **Table S11** (PPF indexes of 3 devices) was included in **Supporting Information S11**: 'PPF index of 3 devices computed in both light and dark condition'.

18 *Page 11, line 270, caption of Figure 3a. it is not clear how the experiment was actually design (how was the light positioned, what was its angle, was the gate electrode obstructing the light, etc.). a picture would be very instructive.*

Page 13, line 322. what would happen if negative pulses were applied in the presence of light?

We thank the reviewer for the comment, we added a schematic of the setup in the Supporting Information.

Furthermore, we performed experiments applying negative gate voltage pulses (between -50 mV and -300 mV) under dark and light conditions (Figure S12.1)

During the first 3 pulses a competing effect in channel current induced by UV light and negative voltage pulses was observed.

Without using light stimulus (Figure S12.1 middle) a current increase proportional to the amplitude of the electric pulse is observed. This behavior can be justified by a backflow of cations from the channel to the electrolyte which causes a reoxidation effect of the PEDOT:PSS, occurring when a negative voltage is applied to the gate. A similar behavior is reported in literature [6].

When the light stimulus is present (Figure S12.1 bottom) the increase of current is less evident during the first three pulses, while increasing the amplitude of the pulse voltages up to -300 mV the electrical contribution is predominant and so the current increase dramatically. Such behavior is probably due to a competition between the effect induced by light (which causes a positive gate polarization and so a current decrease) and those induced by the electric pulses.

Figure S12.1

Changes in the manuscript:

- Inclusion of **Figure S7.3** (schematics of the setup for measurements under light conditions) in **Supporting Information S7**.

Figure S7.3 schematics of set up for measurements under light conditions.

References:

1. Malingappa, P., Thippeswamy, R. & Compton, R. Nitroazobenzene Functionalized Carbon Powder: Spectroscopic Evidence for Molecular Cleavage. *International Journal of Electrochemical Science* **3**, (2008).
2. Goulet-Hanssens, A. *et al.* Hole Catalysis as a General Mechanism for Efficient and Wavelength-Independent Z → E Azobenzene Isomerization. *Chem* **4**, 1740–1755 (2018).
3. Elgrishi, N. *et al.* A Practical Beginner's Guide to Cyclic Voltammetry. *J. Chem. Educ.* **95**, 197–206 (2018).
4. Friedlein, J. T., McLeod, R. R. & Rivnay, J. Device physics of organic electrochemical transistors. *Organic Electronics* **63**, 398–414 (2018).
5. Lan, X. *et al.* p–n hybrid bulk heterojunction enables enhanced photothermoelectric performance with UV–Vis–NIR light. *Nanoscale* **14**, 18003–18009 (2022).
6. Talin, A. A. *et al.* A nonvolatile organic electrochemical device as a low-voltage artificial synapse for neuromorphic computing. *Nature Materials* **16**, 414 (2017).
7. Bonafè, F., Decataldo, F., Fraboni, B. & Cramer, T. Charge Carrier Mobility in Organic Mixed Ionic–Electronic Conductors by the Electrolyte-Gated van der Pauw Method. *Advanced Electronic Materials* **7**, 2100086 (2021).
8. Mariani, F. *et al.* Microscopic Determination of Carrier Density and Mobility in Working Organic Electrochemical Transistors. *Small* **15**, 1902534 (2019).
9. Rivnay, J. *et al.* High-performance transistors for bioelectronics through tuning of channel thickness. *Sci Adv* **1**, e1400251 (2015).
10. Rivnay, J. *et al.* Structural control of mixed ionic and electronic transport in conducting polymers. *Nature Communications* **7**, 11287 (2016).
11. Paudel, P. R., Skowrons, M., Dahal, D., Radha Krishnan, R. K. & Lüssem, B. The Transient Response of Organic Electrochemical Transistors. *Advanced Theory and Simulations* **5**, 2100563 (2022).
12. Zhu, Y. *et al.* Perovskite-Enhanced Silicon-Nanocrystal Optoelectronic Synaptic Devices for the Simulation of Biased and Correlated Random-Walk Learning. *Research (Wash D C)* **2020**, 7538450 (2020).
13. Choi, S., Yang, J. & Wang, G. Emerging Memristive Artificial Synapses and Neurons for Energy-Efficient Neuromorphic Computing. *Adv. Mater.* **32**, 2004659 (2020).
14. Yin, L. *et al.* Optically Stimulated Synaptic Devices Based on the Hybrid Structure of Silicon Nanomembrane and Perovskite. *Nano Lett.* **20**, 3378–3387 (2020).
15. Alquraishi, W. *et al.* Hybrid optoelectronic synaptic functionality realized with ion gel-modulated In₂O₃ phototransistors. *Organic Electronics* **71**, 72–78 (2019).
16. Liu, Z. *et al.* Photoresponsive Transistors Based on Lead-Free Perovskite and Carbon Nanotubes. *Advanced Functional Materials* **30**, 1906335 (2020).

17. He, W. *et al.* A multi-input light-stimulated synaptic transistor for complex neuromorphic computing. *J. Mater. Chem. C* **7**, 12523–12531 (2019).
18. Liu, Y. *et al.* A Hybrid Phototransistor Neuromorphic Synapse. *IEEE Journal of the Electron Devices Society* **7**, 13–17 (2019).
19. Shao, L. *et al.* Optoelectronic Properties of Printed Photogating Carbon Nanotube Thin Film Transistors and Their Application for Light-Stimulated Neuromorphic Devices. *ACS Appl. Mater. Interfaces* **11**, 12161–12169 (2019).
20. Guo, Y. B. *et al.* Low-voltage protonic/photonic synergic coupled oxide phototransistor. *Organic Electronics* **71**, 31–35 (2019).
21. Cheng, W. *et al.* Proton Conductor Gated Synaptic Transistor Based on Transparent IGZO for Realizing Electrical and UV Light Stimulus. *IEEE Journal of the Electron Devices Society* **7**, 38–45 (2019).
22. Wu, Q. *et al.* Photoelectric Plasticity in Oxide Thin Film Transistors with Tunable Synaptic Functions. *Advanced Electronic Materials* **4**, 1800556 (2018).
23. Li, H. K. *et al.* A light-stimulated synaptic transistor with synaptic plasticity and memory functions based on InGaZnOx–Al₂O₃ thin film structure. *Journal of Applied Physics* **119**, 244505 (2016).
24. Yu, J. J. *et al.* Optoelectronic neuromorphic thin-film transistors capable of selective attention and with ultralow power dissipation. *Nano Energy* **62**, 772–780 (2019).
25. Li, B. *et al.* Mimicking synaptic functionality with an InAs nanowire phototransistor. *Nanotechnology* **29**, 464004 (2018).
26. Wang, Y. *et al.* Photonic Synapses Based on Inorganic Perovskite Quantum Dots for Neuromorphic Computing. *Advanced Materials* **30**, 1802883 (2018).
27. Yin, L. *et al.* Synaptic silicon-nanocrystal phototransistors for neuromorphic computing. *Nano Energy* **63**, 103859 (2019).
28. Sun, J. *et al.* Optoelectronic Synapse Based on IGZO-Alkylated Graphene Oxide Hybrid Structure. *Advanced Functional Materials* **28**, 1804397 (2018).
29. Qin, S. *et al.* A light-stimulated synaptic device based on graphene hybrid phototransistor. *2D Mater.* **4**, 035022 (2017).
30. Keene, S. T. A biohybrid synapse with neurotransmitter-mediated plasticity. *Nature Materials* **19**, 16 (2020).
31. Deng, P.-Y. & Klyachko, V. A. The diverse functions of short-term plasticity components in synaptic computations. *Commun Integr Biol* **4**, 543–548 (2011).

List of changes

Table 1: List of changes, manuscript

Page	Line	Changes
4	103-106	Replacement of the sentence: ‘In addition, cyclic voltammetry measurements showed two characteristic peaks (reduction and oxidation peaks at -0.12 V and +0.57 V, respectively) that can be attributed to the reduction of the azobenzene N=N bond induced by the presence of acidic protons in the electrolyte solution ²⁰ (Figure 1d, violet solid line).’ with ‘In addition, cyclic voltammetry measurements showed two characteristic peaks: a reduction peak at -0.12 V that might be attributed to the reduction of the azobenzene N=N bond induced by the presence of protons from -H ₂ PO ₄ in the electrolyte solution and an oxidation peak at +0.57 V which might be due to the oxidation of the azobenzenes.’
4	106	Added the reference: Goulet-Hanssens, A. et al. Hole Catalysis as a General Mechanism for Efficient and Wavelength-Independent Z → E Azobenzene Isomerization. Chem 4, 1740–1755 (2018).
19	515	Addition of pH value of PBS in the sentence: ‘The films’ characterization was performed with an ITO-coated square glass slide (25 mm x 25 mm, Kintec, Hong Kong) as WE, (Ag/AgCl NaCl (3 M) (Redox.me, Sweden) as RE, a platinum wire (Merck Life Science S.r.l., Italy) as CE and PBS solution with pH = 7.4 (100 μL) as supporting electrolyte and applying a voltage from -0.4 V to +1.0 V at a scan rate of 50 mV s ⁻¹ .
20	522	Addition of pH value of PBS in the sentence: ‘The films characterization was performed with an ITO-coated square glass slide as WE, Ag/AgCl NaCl (3 M) (Redox.me, Sweden) as RE and CE and a PBS solution with pH = 7.4 (50 μL). Impedance was measured at +0.01 V and -0.01 V as biased potentials from 105 Hz to 0.1 Hz.’
20	528	Added the sentence: ‘All the measurements have been carried out by using PBS
21	567	(pH = 7.4) as the electrolyte solution.’
20	558	Addition of pH value of PBS in the sentence: ‘The resistor and the azo-OECT have been connected in series to each other by using a tin-based soldering and silver paint; the measurements have been performed using a multichannel setup of a commercial platform (ARKEO, Cicci Research) and PBS (pH = 7.4) as electrolyte solution. ’
21	588	Addition of pH value of PBS in the sentence: ‘The measurement has been performed by connecting two measurement probes to the source and drain electrodes and by connecting a third one to the azo-tz-PEDOT:PSS gate electrode, and by using PBS (pH = 7.4) as electrolyte solution. ’
5	128-129	Replacement of HOMO-x and LUMO+x labeling with the specific HOMO-2 and LUMO+1.
7	148-151	Replacement of caption in Figure 1g: ‘Energy levels diagram of azo-tz-PEDOT:PSS system obtained from DFT/TDDFT calculations, the arrow shows the possible photo/induced electron transfer from the LUMO level (localized on the photoexcited azobenzene) to the HOMO level of the system (localized on the PEDOT backbone).’ with ‘Chemical structure of the azo-tz-PEDOT system and energy levels of cis conformer obtained from DFT/TDDFT calculations. Only HOMO -2/LUMO levels localized on azo-tz (pink box) and HOMO/LUMO +1 localized on PEDOT (blue box) are displayed. The arrow shows the possible photoinduced electron transfer from the LUMO level (localized on the photoexcited azobenzene) to the HOMO level of the system (localized on the PEDOT backbone).’
5	122-134	Replacement of the sentence: ‘To support this hypothesis, theoretical investigations were carried out by means of ab initio calculations. An effective model of the azo-tz-PEDOT system was provided through DFT calculation (Experimental Section) to derive the energy diagram of trans and cis isomers for

the azo-tz-PEDOT system (i.e., HOMO and HOMO-x). The corresponding vertical excitation energy obtained from TDDFT allowed to calculate the energy values of unoccupied states (i.e., LUMO and LUMO+x) (Supporting Information S8). Here, the levels alignment is not affected by the conformational orientation, suggesting that the cis/trans isomerization is not a key process in the charge transfer mechanism (Figure 1g). However, it was clearly shown that intramolecular charge transfers may take place upon photoexcitation leading to spatially separated charge carriers that, in principle, can be trapped on the azo side inducing a polarization of the gate electrode as a result of photoinduced charge trapping.’ **With** ‘DFT/TD-DFT calculations were performed on a azo-tz-PEDOT model system to support the interpretation of the light-responsive mechanisms and to provide an unbiased energetic picture of the possible light-induced charge transfer processes (see Section S8 in SI for method validation). Our results confirmed the nature of the UV–vis transitions for cis and trans conformers as found in experiments (see Fig. S8.3 and Fig. 1c for comparison). For the cis conformer, we predicted the following energy diagram: the HOMO is localized on the PEDOT backbone, the LUMO is instead localized on the azo-tz moiety; the other relevant MO is the HOMO-2 that is localized onto the azo-tz moiety (see Fig. 1g). Within this energy framework, upon UV light irradiation, the photoexcited electron can be transferred from the LUMO (on the azo-tz) to the HOMO on the PEDOT backbone conductive system. Population of states located on the azobenzenes would lead to spatially separated charge carriers that, in principle, can be trapped on the azo side (positive charge highlighted in Fig. 1g). This mechanism would clarify the polarization at the gate electrode and the subsequent decrease in the current flow.’

6 **Refinement of Figure 1g.**

22 607-610 **Added the section “Data Analysis”, as follow:** “All the numerical data analysis was carried through custom-made MATLAB scripts. All the current (conductance) modulations were computed as a percentage with respect to each device baseline. Statical analysis was conducted on current (conductance) variations expressed in percentages.”

Update of Figure 1f to fix the mismatch between the points of the x-axis and the y-axis.

7 156-162 **Replacement of the sentence:** “As such, the resulting charges flowing to the channel caused a dedoping of the PEDOT:PSS channel and a related decrease of I_{ds} .” **with:** “Such positive charge accumulation at the gate electrode, induced by the UV light irradiation, resulted in an effective gating of the transistor. As a consequence, positive charged ions can penetrate the bulk of the polymeric channel, causing the dedoping of the PEDOT:PSS and the resulting decrease of I_{ds} .”.

Revision of letters in Figure 2 which is also provided in high resolution.

Revision of label in Figure 2gii which is also provided in high resolution.

5-6 118-122 **Addition in the text as follows:** ‘Interestingly, under light exposure ($t = 5$ minutes, $\lambda = 365$ nm, 6 W), the I_{ds} current shifts to lower values suggesting a photoinduced charge transfer/charge trapping mechanism occurring between the azobenzenes and the PEDOT:PSS backbone (Figure 1g), *unlike of PEDOT:PSS alone which typically does not show photothermoelectrical behavior* [ref].’

6 122 **Inclusion of the reference in the text:** Lan, X. *et al.* p–n hybrid bulk heterojunction enables enhanced photothermoelectric performance with UV–Vis-NIR light. *Nanoscale* 14, 18003–18009 (2022).

12-13 314-318 **Replacement of the sentence:** ‘In fact, when a different electrolyte (i.e., an aqueous solution 100 mM of NaCl), containing two orders of magnitude less protons compared to PBS, was used, the application of consecutive negative voltage pulses induced a very poor modulation of I_{ds} confirming that in the

- variation of the current values depends on the amount of protons available in the electrolyte³² (Supporting information S12).’ **with** ‘In fact, when a different electrolyte (i.e., an aqueous solution 100 mM of NaCl), containing two orders of magnitude less protons compared to PBS, was used, the application of consecutive negative voltage pulses induced a very poor modulation of Ids confirming that in the variation of the current protons in the electrolyte are involved in the mechanism 32 (Supporting information S12).’
- 12 298 **Changed the expression** “charge mobility” to “ions migration”.
- 3 53-59 **Revision of the abstract**
Revision of the text: ‘However, traditional inorganic and hybrid optoelectronic platforms find major limitations in biologically driven systems where ion sensing as well as tailored biomechanical properties are required^{7,11}. Thereby, in order to fabricate multimodal platforms capable to mimic a potentially also directly interface biological systems such as neuronal cells, several strategies have been proposed for the development of optoelectronic devices featuring electromechanical stability in aqueous environment, ionic-electronic transduction and biophysical properties (i.e., biocompatibility, low stiffness¹³⁻¹⁵).’
- 3 73-77 **Revision of the text:** Interestingly, this newly synthesized light-responsive and its integration conjugated polymer and its integration as gate electrode of this neurohybrid building block allowed to achieve and “all-in-one” device capable of both biomimic vertebrate retina visual pathways and to operate as an optical neuromorphic synaptic device capable to emulate both short and long term plasticity (STP and LTP).’ ...
- 7 174 **Removed the phrasing** “a more complex”.
Added the sentence: ‘Furthermore, the gating efficiency of the device could be finely tuned, by optimizing the geometrical ratio ($\frac{Wt}{L}$) of the OPECT channel, increasing the ON/OFF ratio of the device, while decreasing the response time.’
- 15 370-372 **Added the sentence:** ‘In perspective, by optimizing the geometrical ratio ($\frac{Wt}{L}$) of the OPECT channel, the light-mediated gating efficiency could be finely tuned, also reducing the switching speed of the device.’
- 10-11 262-264 **Replacement of the sentence:** ‘...suggesting that electrical stimuli elicit STP in the neuromorphic azo-OECT.’ **with** ‘PPF indexes calculated under dark and light conditions are comparable, suggesting that STP is independent of the light exposure and mainly induced by the electrical stimulation.’
Added ITO symbol in the legend of Figure 1a.
- 17 439 **Addition of the sentence as follows:** ‘Organic electrochemical transistor fabrication. Customized patterned ITO coated glass (Surface resistivity 20 Ω cm⁻², 25 mm x 25 mm, Kintec, Hong Kong), *see Figure S7.2*, were cleaned in an ultrasonic bath with Alconox® detergent solution (Merck Life Science S.r.l., Italy), deionized water, acetone (Merck Life Science S.r.l., Italy), and 2-propanol (Merck Life Science S.r.l., Italy) (10 minutes for each solvent) and dried with compressed air.’
- 18 455-460 **Replacement of the sentence:** ‘Finally, to remove the excess of PEDOT:PSS and fabricate the OECT channel, a plasma etching (15 minutes, 100 W) has been performed covering both the planar gate and the channel with two PDMS masks (15 mm x 7 mm, 23 mm x 12 m). Before the use, the device was gently rinsed in deionized water overnight to promote the PEDOT:PSS swelling.’ **with** ‘Finally, to remove the excess of PEDOT:PSS and pattern the OECT channel, a plasma etching (15 minutes, 100 W) was performed including a hard PDMS mask (15 mm x 7 mm, 23 mm x 12 mm) to cover both the planar gate and the channel with two PDMS masks. The so obtained OECTs have a channel area of 15 mm x 7 mm, a gate area of 13 mm x 10 mm of which the illuminated area is 20 mm².

		Before the use, the device was gently rinsed in deionized water overnight to promote the PEDOT:PSS swelling.'
7	165-168	Replacement of the sentence: ' Indeed, the spontaneous cis-trans relaxation occurs in the order of minutes in similar azobenzene monomers ²⁵ , while here the current decrease observed in the azo-OECT is almost instantaneous, suggesting that the cis-trans isomerization contribution can only have an influence within the first light pulse ^{25,26} .' with 'Indeed, the spontaneous cis-trans relaxation occurs in the order of minutes in similar azobenzene monomers ²⁵ , while here the current decrease observed in the azo-OECT is in the order of seconds (10-30s), reaching the initial current values. This suggest that the cis-trans isomerization contribution can only have an influence within the first light pulse ^{25,26} .'
8	185	Revision of the text: 'In the light of this, the retina cell layers'
10	228-231	Revision of the text: 'Indeed, as previously shown ³⁵ , platform can mimic neuromorphic features when square voltage pulses are applied at the gate terminal, acting as action potentials at presynaptic ends in biological neuronal synapses and modulating the conductance at the channel (post-synaptic end) ³⁵ (Figure 3a).'
10	228	Inclusion of the reference: Keene, S. T. A biohybrid synapse with neurotransmitter-mediated plasticity. Nature Materials 19, 16 (2020).
10	234-235	Revision of the text: 'decrease in the conductance of the postsynaptic channel was observed'...

Table 2: List of changes, supporting information

Section	Changes
S11	Added Table S11 to show PPF indexes extracted from N=3 samples.
S8	Inclusion of Figure S8.4 in Supporting Information S8.4
S6	Added new labels in Figure S6a and Figure S6b in Supporting Information S6 to identify the semicircular shape of PEDOT:PSS at high frequencies and of azo-tz-PEDOT:PSS at low frequencies
S10	Added Figure S10.3. Splitted Table S10 in Table S10.1 and S10.2 in Supporting Information S10.
S10	Addition of the caption of Figure S10.3: 'Conditioning calculated as percentage conductance variation ($\Delta G\%$) by changing the light intensity of the UV light (20% , 60%, 100%, N = 3 of each) and the PW applied to the gate electrode (500 milliseconds, 1 second, 3 seconds, 5 seconds and 8 seconds, N = 3 of each).'
S10	Addition of the caption of Table S10.1: 'Conditioning calculated as percentage conductance variation ($\Delta G\%$) by changing the light intensity of the UV light (20% , 60%, 100%, N = 3 of each) and the PW applied to the gate electrode (500 milliseconds, 1 second, 3 seconds, 5 seconds and 8 seconds, N = 3 of each).'
S10	Addition of the caption of Table S10.2: 'Extinction calculated as percentage conductance variation ($\Delta G\%$) by changing the light intensity of the UV light (20% , 60%, 100%, N = 3 of each) and the PW applied to the gate electrode (500 milliseconds, 1 second, 3 seconds, 5 seconds and 8 seconds, N = 3 of each).'
S7	Inclusion of ITO glass substrate schematics as Figure S7.2 in Supporting Information S7.
S7	Figure S7 has been renamed as Figure S7.1 in Supporting Information S7.

S7

Inclusion of Figure S7.3 (schematics of the setup for measurements under light conditions) in Supporting Information S7.

REVIEWER COMMENTS

Reviewer #1 (Remarks to the Author):

Some concerns still remain:

Major issue:

1. With different numbers of 365 nm light pulse, the gate comprised different proportions of trans/cis isomer and different redistribution of energy diagram, probably with different charge transfer barriers and thus different photo to electric conversion efficiency. Did the gate possess the same gating effect upon the optical pulse regardless of the different trans/cis proportions? Or only the cis gate was used to gate the channel? Experimental details are recommended and the conclusion should be stated in the text.
2. By saying the “saturated modulation” of the azo-tz gate, we noticed that, this gate could induce I_{ds} change of ca. 0.005 mA at 100 mV gate pulse in Figure 4b, and this value was kept at 120 mV pulse. But in Figure S12.3, using the PEDOT:PSS gate, the change of I_{ds} stepwise increased as the gate pulse increased to 300 mV and did not tend to be stable. What caused these differences? Also, in Figure S12.3 under UV light, why the positive gate pulse could cause the baseline elevation and the negative pulse could not?
3. The authors still did not explain the impact of the baseline level on channel currents when a stimulus was exerted. Different baselines in many figures, including but not limited to -9×10^{-5} A in Figure 2e (ii), -2.85×10^{-4} A in Figure 3f, and -0.170 mA in Figure 4b. Did the baseline level influence the I_{ds} change upon the electric or the light pulse? How?
4. No doubt about the charge transfer from the azo-tz to the PEDOT:PSS. But whether the gate current originated from the capacitive one, or the faradic one, or both, should be discussed in detail. If no Faraday current occurred at the gate, why did the gate current in Figure 2e stabilize at a certain value without significant decay after illumination?

Minor issue:

1. In Page S11, the author claimed that the PEDOT:PSS “at high frequencies show a semicircular profile”. No semicircular shape could be observed in Figure S6a.
2. In Page 12, line 292, the author claimed that 500 light pulses were applied as shown in Figure 4a, but only one light pulse could be seen.
3. Still, in Figure 1f, the output curves did not pass the zero point.
4. The authors should carefully check the labels in the Figures, especially in Figure 2 and the corresponding text and the captions.
5. Same units are recommended for the same physical parameter.

Reviewer #2 (Remarks to the Author):

The author has made corresponding modifications according to the suggestions and recommends publication.

Reviewer #3 (Remarks to the Author):

In general, the reviewer is happy with changes / corrections made.

A few minor comments pertain to grammar / style...

In Abstract, is: "... towards a multimodal devices...". Should be either "... towards a multimodal device" or "... towards multimodal devices.

In Introduction, 2nd paragraph, is: "... multimodal platforms capable TO mimic...". Should/could be "... multimodal platforms capable OF mimicking [...] directly interfACING WITH biological...".

In Conclusion, 2nd paragraph, "... exhibited IDEAL neuromorphic features...". Please consider the use of "ideal" here (at best it's speculative, at worst it's meaningless).

Reviewers' Comments to the manuscript

Reviewer #1:

Some concerns still remain:

Major issue:

- 1. With different numbers of 365 nm light pulse, the gate comprised different proportions of trans/cis isomer and different redistribution of energy diagram, probably with different charge transfer barriers and thus different photo to electric conversion efficiency. Did the gate possess the same gating effect upon the optical pulse regardless of the different trans/cis proportions? Or only the cis gate was used to gate the channel? Experimental details are recommended and the conclusion should be stated in the text.*

We thank the reviewer for the comment. Actually, with a variable number of light pulses (at 365 nm), the gate should comprise the majority of cis isomers considering that the transition from trans-cis occurs in the range of nanoseconds and the reverse isomerization from cis-to-trans in the range of minutes (as demonstrated in the Supporting Information S5)¹. For this reason, when multiples pulses are applied, we consider that the only isomer involved is the cis one because in the short interval of dark it does not have time to reverse the conformation. The differences in the electrical current observed are indeed ascribable to the photoexcitation of the azo-tz-PEDOT system independently of the isomer involved. However, even considering different proportions of trans/cis isomer at the gate, the theoretical calculations showed that the energy diagram of frontier orbitals is not qualitatively affected by the different isomer conformations. So different trans/cis proportions should not affect the accumulation of the charge on the gate electrode, because for both the trans and cis isomers the analysis of MO energies and localisations shows the possibility of a convenient and direct injection for a photoexcited electron from the LUMO level of the azobenzenes to the HOMO level of PEDOT. Furthermore, the induced charge calculated by applying 20 short light pulses (PW = 2 seconds, time interval 10 seconds) remains constant during each pulse, as shown in the figure below, confirming the gating effect is mainly due to the photoinduced charge transfer in the Azo-PEDOT is independent cis-trans portions, of present.

Changes in the text: in the supporting as last sentence of the S8 section: “From the above results emerges that MO energies and localizations for cis and trans isomers are both suitable for direct injection for a photoexcited electron from the LUMO level of the azobenzenes to the HOMO level of PEDOT. This means that even considering a not complete conversion from trans/cis isomer at the gate, the energy diagram of frontier orbitals is not qualitatively affected by the different isomer conformations.”

2. *By saying the “saturated modulation” of the azo-tz gate, we noticed that, this gate could induce I_{ds} change of ca. 0.005 mA at 100 mV gate pulse in Figure 4b, and this value was kept at 120 mV pulse. But in Figure S12.3, using the PEDOT:PSS gate, the change of I_{ds} stepwise increased as the gate pulse increased to 300 mV and did not tend to be stable. What caused these differences? Also, in Figure S12.3 under UV light, why the positive gate pulse could cause the baseline elevation and the negative pulse could not?*

We thank the reviewer for the comment. The visual difference between the measurements shown in Figure 4b and Figure S12.3 is mainly because in the former the voltage was stepped of 20 mV per pulse, while in the latter the increase in the gate voltage was 50 mV per pulse. For this reason, the change between 100 mV and 120 mV shown in Figure 4b is not easily noticeable. Regarding the second part of this comment, the slight baseline current increase in Figure S12.3 can be ascribed to ionic drift that is typical of OECTs. As the drain electrode is biased with a negative voltage ($V_{DS} = -0.2$ V), the application of a positive bias at the gate terminal ($V_{GS} \geq 0$ V) allows for a slow ionic migration from the electrolyte to the PEDOT:PSS channel. On the other hand, the negative gate bias does not allow for cations to migrate inside the polymeric channel, as they will migrate towards the gate electrode². In addition, control measurements were in fact carried out to check whether the UV light could induce some gating effect also in case of pristine PEDOT:PSS. As shown in the figure, no gate current can be recorded when shining light on bare PEDOT:PSS.

3. *The authors still did not explain the impact of the baseline level on channel currents when a stimulus was exerted. Different baselines in many figures, including but not limited to -9×10^{-5} A in Figure 2e (ii), -2.85×10^{-4} A in Figure 3f, and -0.170 mA in Figure 4b. Did the baseline level influence the I_{ds} change upon the electric or the light pulse? How?*

We thank the reviewer for the comment. The baseline level of the channel current is an intrinsic property of the channel of the transistor, *i.e.*, it is completely independent of the gate, as it describes the electrical conductivity of the PEDOT:PSS film that connects drain and source electrodes. For this reason, the baseline level of the current does not impact the gating of the device (either light induced or electrical gating). Such difference among different devices can be ascribed to the non-ideal behaviour of conducting polymer thin layers³. On the other hand, such dispersion hampers a direct analysis of the gating effect, as it happens in different current ranges (from few to tens μ A). To reduce the effect of such baseline current dispersion and to perform a statistically relevant analysis, all the conductance variations were

computed as percentages, *i.e.*, for each device the baseline current represent the 100% of the total current and the variation is computed with regards to this baseline level. Statistical analysis (mean and s.d.) is then performed using such percentage values, so that it is possible to compare the light-induced and the electrical-induced effects among different devices.

No doubt about the charge transfer from the azo-tz to the PEDOT:PSS. But whether the gate current originated from the capacitive one, or the faradic one, or both, should be discussed in detail. If no Faraday current occurred at the gate, why did the gate current in Figure 2e stabilize at a certain value without significant decay after illumination?

We thank the reviewer for the comment. OECTs rely on the capacitive coupling between gate and electrolyte to efficiently dope/de-dope the transistor channel². In addition, the ratio between gate and channel capacitance is essential in biosensing application as the voltage drop at gate/electrolyte and channel/electrolyte interfaces depends on the capacitive divider between the two². At the same time, PEDOT:PSS-gated OECTs can achieve high sensitivity of faradic currents through trans-conductance modulation, as the conducting polymer dampens the capacitive coupling between gate and electrolyte, with respect to ion impermeable electrodes (Au or similar)⁴. In our work, both capacitive and faradic currents are present. In case of voltage gating, capacitive coupling typical of OECTs occurs. In case of light, a faradic current can be observed, as it does not decay in time (Figure 2e of the manuscript). Such faradic mechanism agrees with the proposed light-induced mechanism, as follows: upon light exposure, electrons are transferred from the LUMO (azo-tz) to the HOMO (PEDOT backbone), while holes are kept in the azo-tz. As a result, the GATE/electrolyte interface potential decreases inducing a faradic current in the electrolyte, *de facto* increasing the channel/electrolyte potential, de-doping the polymeric channel⁵.

Changes in the text: (page 6): “Notably, while electrical gating induces a capacitive gate current, the light exposure generates a faradic gate current, as charges are transferred from the gate to the electrolyte, during the whole light exposition, without showing a significant decay (**Figure 2e-ii**). Such current generation is in agreement with the light-induced mechanisms previously proposed (**Figure 1g**): upon light exposure, electrons are transferred from the LUMO (azo-tz) to the HOMO (PEDOT backbone), while holes are kept in the azo-tz. As a result, the GATE/electrolyte interface potential decreases inducing a faradic current in the electrolyte, thus increasing the channel/electrolyte potential, de-doping the polymeric channel⁵”.

Minor issue:

1. *In Page S11, the author claimed that the PEDOT:PSS “at high frequencies show a semicircular profile”. No semicircular shape could be observed in Figure S6a.*

We thank the reviewer for the comment. Supporting Information S6 (Page S11) was modified, as the previous version was confusing for the reader. Paragraph S6 is now updated, correctly describing the semicircular shape only in case of azo-tz-PEDOT:PSS (Figure S6b).

2. *In Page 12, line 292, the author claimed that 500 light pulses were applied as shown in Figure 4a, but only one light pulse could be seen.*

We thank the reviewer for the comment. In figure 4a 500 light pulses were applied continuously stimulating the azo-tz-PEDOT:PSS. The duration of each pulse was set to 1 s, and the delay between pulses was set to 1 s. For this reason, given the time scale, it is not possible to distinguish the effect of the single pulses. A zoomed plot was added in the Supporting Information S13 (Page S29) to allow for a better visualization of the pulses.

3. *Still, in Figure 1f, the output curves did not pass the zero point.*

We thank the reviewer for the comment. One of the transcharacteristic curves does not pass the zero point that is ascribable to a leakage current induced by the voltage applied at the gate terminal, because of minor fabrication defects. Indeed, the curve that is not passing the zero point (both before and after UV exposition) is sampled while the gate electrode is biased with +800 mV, meaning that the higher the gate voltage, the higher the leakage current. Still, such leakage does not affect the performance/operation of the device.

4. *The authors should carefully check the labels in the Figures, especially in Figure 2 and the corresponding text and the captions.*

We thank the reviewer for the comment. All labels and captions were carefully revised and updated

5. *Same units are recommended for the same physical parameter.*

We thank the reviewer for the comment. The whole manuscript was updated, and now same units are used for same physical parameters.

Reviewer #3:

In general, the reviewer is happy with changes / corrections made.

A few minor comments pertain to grammar / style...

1. *In Abstract, is: "... towards a multimodal devices...". Should be either "... towards a multimodal device" or "... towards multimodal devices.*
2. *In Introduction, 2nd paragraph, is: "... multimodal platforms capable TO mimic...". Should/could be "... multimodal platforms capable OF mimicking [...] directly interfacing WITH biological...".*
3. *In Conclusion, 2nd paragraph, "... exhibited IDEAL neuromorphic features...". Please consider the use of "ideal" here (at best it's speculative, at worst it's meaningless).*

We thank the reviewer for the comments that have been all addressed.

References

1. Garcia-Amorós, J., Martínez, M., Finkelmann, H. & Velasco, D. Kinetic-Mechanistic Study of the Thermal Cis-to-Trans Isomerization of 4,4'-Dialkoxyazoderivatives in Nematic Liquid Crystals. *J. Phys. Chem. B* **114**, 1287–1293 (2010).
2. Friedlein, J. T., McLeod, R. R. & Rivnay, J. Device physics of organic electrochemical transistors. *Organic Electronics* **63**, 398–414 (2018).
3. Inzelt, G. & Láng, G. G. Electrochemical Impedance Spectroscopy (EIS) for Polymer Characterization. in *Electropolymerization* 51–76 (John Wiley & Sons, Ltd, 2010). doi:10.1002/9783527630592.ch3.
4. Gualandi, I. *et al.* Selective detection of dopamine with an all PEDOT:PSS Organic Electrochemical Transistor. *Sci Rep* **6**, 35419 (2016).
5. Xie, K. *et al.* Organic electrochemical transistor arrays for real-time mapping of evoked neurotransmitter release in vivo. *Elife* **9**, (2020).

List of changes

In this revision process, the authors have been supported by an overall language editing and language proof throughout the whole manuscript.

Table 1: List of changes, manuscript

Page	Line	Changes
6		Notably, while electrical gating induces a capacitive gate current, the light exposure generates a faradic gate current, as charges are transferred from the gate to the electrolyte, during the whole light exposition, without showing a significant decay (Figure 2e-ii). Such current generation is in agreement with the light-induced mechanisms previously proposed (Figure 1g): upon light exposure, electrons are transferred from the LUMO (azo-tz) to the HOMO (PEDOT backbone), while holes are kept in the azo-tz. As a result, the GATE/electrolyte interface potential decreases inducing a faradic current in the electrolyte, thus increasing the channel/electrolyte potential, de-doping the polymeric channel ⁵ .

Table 2: List of changes, supporting information

Section	Changes
S8	Added: "From the above results emerges that MO energies and localizations for cis and trans isomers are both suitable for direct injection for a photoexcited electron from the LUMO level of the azobenzenes to the HOMO level of PEDOT. This means that even considering a not complete conversion from trans/cis isomer at the gate, the energy diagram of frontier orbitals is not qualitatively affected by the different isomer conformations."
S6	Improved the description of the Nyquist plot.
S13	Added a zoom-in of Figure 4a to allow for a better visualization.

REVIEWERS' COMMENTS

Reviewer #1 (Remarks to the Author):

The authors have addressed my comments and I believe the revised manuscript is in good shape to be published.

Reviewer #3 (Remarks to the Author):

The reviewer is happy with all of the changes and corrections made, and recommends the article for publication.